# Dimension-Free Adaptive Subgradient Methods with Frequent Directions

**Sifan Yang** [* 1 2]  **Yuanyu Wan** [* 3 4]  **Peijia Li** [1 2]  **Yibo Wang** [1 2]  **Xiao Zhang** [5]  **Zhewei Wei** [5]  **Lijun Zhang** [1 6 2]

## Abstract

In this paper, we investigate the acceleration of adaptive subgradient methods through frequent directions (FD), a widely-used matrix sketching technique. The state-of-the-art regret bound exhibits a *linear* dependence on the dimensionality $d$, leading to unsatisfactory guarantees for high-dimensional problems. Additionally, it suffers from an $O(\tau^2 d)$ time complexity per round, which scales quadratically with the sketching size $\tau$. To overcome these issues, we first propose an algorithm named FTSL, achieving a tighter regret bound that is independent of the dimensionality. The key idea is to integrate FD with adaptive subgradient methods under *the primal-dual framework* and add the cumulative discarded information of FD back. To reduce its time complexity, we further utilize fast FD to expedite FTSL, yielding a better complexity of $O(\tau d)$ while maintaining the same regret bound. Moreover, to mitigate the computational cost for optimization problems involving matrix variables (e.g., training neural networks), we adapt FD to Shampoo, a popular optimization algorithm that accounts for the structure of decision, and give a novel analysis under *the primal-dual framework*. Our proposed method obtains an improved dimension-free regret bound. Experimental results have verified the efficiency and effectiveness of our approaches.

---

[*]Equal contribution [1]National Key Laboratory for Novel Software Technology, Nanjing University, Nanjing, China [2]School of Artificial Intelligence, Nanjing University, Nanjing, China [3]School of Software Technology, Zhejiang University, Ningbo, China [4]Hangzhou HighTech Zone (Binjiang) Institute of Blockchain and Data Security, Hangzhou, China [5]Gaoling School of Artificial Intelligence, Renmin University of China, Beijing, China [6]Pazhou Laboratory (Huangpu), Guangzhou, China. Correspondence to: Lijun Zhang <zhanglj@lamda.nju.edu.cn>.

*Proceedings of the 42$^{nd}$ International Conference on Machine Learning*, Vancouver, Canada. PMLR 267, 2025. Copyright 2025 by the author(s).

## 1. Introduction

Adaptive subgradient methods have attracted considerable research interest in past decades, which simplify the learning rate selection while ensuring that their regret bounds are comparable to those obtained through manual tuning (Duchi et al., 2010a; Hazan & Koren, 2012; Agarwal et al., 2019). The pioneering work of Duchi et al. (2011) introduces adaptive subgradient methods with full matrices (ADA-FULL) within both the primal-dual subgradient framework (Xiao, 2009) and the mirror descent framework (Duchi et al., 2010b). ADA-FULL achieves a regret bound of $O(\text{tr}(G_T^{1/2}))$, where $G_T$ is the sum of gradient outer products over $T$ rounds, and this regret bound is better than that of non-adaptive methods when gradients are sparse. However, ADA-FULL requires maintaining a preconditioning matrix to store the past gradient outer products and computing the inverse of this preconditioning matrix, resulting in an $O(d^2)$ space complexity and an $O(d^3)$ time complexity, respectively, where $d$ is the dimensionality. Thus, ADA-FULL is impractical for large-scale machine learning tasks involving high-dimensional data.

To address these limitations, several studies propose adopting frequent directions (FD) (Ghashami et al., 2016) to reduce the computational complexity of ADA-FULL (Wan et al., 2018; Wan & Zhang, 2022; Feinberg et al., 2023). In particular, Wan et al. (2018) first develop an efficient variant of ADA-FULL, namely ADA-FD, by employing FD to approximate the sum of gradient outer products over the past rounds. Let $\tau \ll d$ denote the sketching size and $\rho_t$ denote the discarded eigenvalue of FD in round $t$. ADA-FD reduces the space and time complexities to $O(\tau d)$ and $O(\tau^2 d)$, while enjoying $O(\text{tr}(G_T^{1/2}) + \sum_{t=1}^{T} \sqrt{\rho_t})$ and $O(\text{tr}(G_T^{1/2}) + \sum_{t=1}^{T} \tau\sqrt{\rho_t})$ regret bounds under the primal-dual subgradient framework and the mirror descent framework, respectively. Moreover, by exploiting an accelerated trick for FD, Wan & Zhang (2022) further propose ADA-FFD with $O(\tau d)$ space and time complexities, while keeping the same regret bounds. Recently, Feinberg et al. (2023) design Sketchy-ADAGRAD (S-ADA) under the mirror descent framework, which achieves a better $O(\text{tr}(G_T^{1/2}) + \sqrt{d(d-\tau)\rho_{1:T}})$ regret bound, where $\rho_{1:T} = \sum_{t=1}^{T} \rho_t$. The key idea is to utilize a variant of FD (Chen et al., 2020), which adds back the cumulative

*Table 1.* Comparison of ADA-FULL and its FD-based variants, where ADA-FD (P) and ADA-FD (M) represent ADA-FD under the primal-dual subgradient framework the mirror descent framework, respectively. We denote $\lambda_i$ be the $i$-th eigenvalue of $G_T$.

| Algorithms | Regret Bounds | Space | Time |
|---|---|---|---|
| ADA-FULL (Duchi et al., 2011) | $O(\text{tr}(G_T^{1/2}))$ | $O(d^2)$ | $O(d^3)$ |
| ADA-FD (P) (Wan et al., 2018) | $O(\text{tr}(G_T^{1/2}) + \sum_{t=1}^{T} \sqrt{\rho_t})$ | $O(\tau d)$ | $O(\tau^2 d)$ |
| ADA-FD (M) (Wan et al., 2018) | $O(\text{tr}(G_T^{1/2}) + \sum_{t=1}^{T} \tau\sqrt{\rho_t})$ | $O(\tau d)$ | $O(\tau^2 d)$ |
| S-ADA (Feinberg et al., 2023) | $O(\text{tr}(G_T^{1/2}) + \sqrt{d(d-\tau)\rho_{1:T}})$ | $O(\tau d)$ | $O(\tau^2 d)$ |
| **FTSL** (this work) | $O(\text{tr}(G_T^{1/2}) + \sqrt{\rho_{1:T}})$ | $O(\tau d)$ | $O(\tau^2 d)$ |

*Table 2.* Comparison of FFD-based variants of ADA-FULL, where ADA-FFD (P) and ADA-FFD (M) represent ADA-FFD under the primal-dual subgradient framework and the mirror descent framework, respectively.

| Algorithms | Regret Bounds | Space | Time |
|---|---|---|---|
| ADA-FFD (P) (Wan & Zhang, 2022) | $O(\text{tr}(G_T^{1/2}) + \sum_{t=1}^{T} \sqrt{\rho_t})$ | $O(\tau d)$ | $O(\tau d)$ |
| ADA-FFD (M) (Wan & Zhang, 2022) | $O(\text{tr}(G_T^{1/2}) + \sum_{t=1}^{T} \tau\sqrt{\rho_t})$ | $O(\tau d)$ | $O(\tau d)$ |
| **Fast S-ADA** (this work) | $O(\text{tr}(G_T^{1/2}) + \sqrt{d(d-\tau)\rho_{1:T}})$ | $O(\tau d)$ | $O(\tau d)$ |
| **FTFSL** (this work) | $O(\text{tr}(G_T^{1/2}) + \sqrt{\rho_{1:T}})$ | $O(\tau d)$ | $O(\tau d)$ |

dropped eigenvalues (referred to as the escaped mass) to keep the positive definite monotonicity of the preconditioning matrix. However, its regret bound depends on $d$, leading to unsatisfactory guarantees for high-dimensional problems, and its time complexity is $O(\tau^2 d)$, which is worse than that of ADA-FFD. Thus, it is natural to ask *whether the regret bound and the time complexity of Feinberg et al. (2023) can be further improved*.

In this paper, we provide an affirmative answer to this question. Specifically, we first develop an algorithm, namely **F**ollow-**t**he-**S**ketchy-**L**eader (FTSL), to enhance the existing regret bound. We integrate FD with ADA-FULL under *the primal-dual framework* and add the cumulative discarded eigenvalues of FD back. FTSL enjoys a tighter *dimension-free* $O(\text{tr}(G_T^{1/2}) + \sqrt{\rho_{1:T}})$ regret bound, while obtaining the space and time complexities of $O(\tau d)$ and $O(\tau^2 d)$. Additionally, we propose an accelerated variant of FTSL, named FTFSL, by doubling the sketching size to reduce the number of time-consuming computations. FTFSL preserves the regret bound and space complexity of FTSL, while simultaneously lowering the time complexity to $O(\tau d)$ when $\tau \leq \sqrt{d}$. Remarkably, we can also improve the time complexity of S-ADA by using this technique, but its regret bound still remains inferior to that of FTFSL. We summarize our results and comparisons with the previous work in Table 1 and Table 2.

Moreover, we investigate optimization problems with matrix variables $X_t \in \mathbb{R}^{m \times n}$, a scenario commonly encountered in deep learning tasks. In this case, one can apply the aforementioned methods by flattening the gradient $G_t^X \in \mathbb{R}^{m \times n}$ into a vector $\mathbf{g}_t \in \mathbb{R}^{mn}$, which, however, incurs a memory usage of $O(\tau mn)$. To improve the memory efficiency, Feinberg et al. (2023) have adapted FD to Shampoo (Gupta et al., 2018), a popular adaptive preconditioning method that accounts for the structure of the parameters. Although Feinberg et al. (2023) reduce the space complexity to $O(\tau(m + n))$, their regret bound again relies on the dimensionality $m, n$. To address this issue, we integrate FD with a primal-dual variant of Shampoo and obtain a dimension-free regret bound via a novel analysis. Our approach, termed FTSL-Shampoo, attains an enhanced theoretical guarantee that is independent of the dimensionality $m, n$. We contrast FTSL-Shampoo with previous methods in Table 3. Finally, we conduct experiments on online classification and neural network training to validate the superiority of our methods.

## 2. Related Work

In this section, we briefly review the related work on adaptive subgradient methods, their fast variants based on sketching, and Shampoo.

*Table 3.* Comparison of adaptive subgradient methods for the case where the decision has a matrix structure $X_t \in \mathbb{R}^{m \times n}$. We denote the gradient $G_t^X \in \mathbb{R}^{m \times n}$, $r$ is the largest rank of $G_t^X$, $L_T = \epsilon I_{m \times m} + \sum_{t=1}^T G_t^X (G_t^X)^\top$, $R_T = \epsilon I_{n \times n} + \sum_{t=1}^T (G_t^X)^\top G_t^X$, $\epsilon$ is a hyper-parameter, $\rho_{1:T}^L$ and $\rho_{1:T}^R$ represent the sum of the removed eigenvalues in FD during the approximation of $\sum_{t=1}^T G_t^X (G_t^X)^\top$ and $\sum_{t=1}^T (G_t^X)^\top G_t^X$, respectively. For ADA-FULL and FTFSL, we define $G_T = \sum_{t=1}^T \mathbf{g}_t \mathbf{g}_t^\top \in \mathbb{R}^{mn \times mn}$, where $\mathbf{g}_t = \overline{\text{vec}}(G_t^X) \in \mathbb{R}^{mn}$ and $\overline{\text{vec}}(\cdot)$ denotes the row-major vectorization of a matrix.

| Algorithms | Regret Bounds | Space | Time |
|---|---|---|---|
| ADA-FULL (Duchi et al., 2011) | $O(\text{tr}(G_T^{1/2}))$ | $O(m^2 n^2)$ | $O(m^3 n^3)$ |
| **FTFSL** (this work) | $O(\text{tr}(G_T^{1/2}) + \sqrt{\rho_{1:T}})$ | $O(\tau mn)$ | $O(\tau mn)$ |
| Shampoo (Gupta et al., 2018) | $O(\sqrt{r}\,\text{tr}(L_T^{1/4})\,\text{tr}(R_T^{1/4}))$ | $O(m^2 + n^2)$ | $O(m^3 + n^3)$ |
| S-Shampoo (Feinberg et al., 2023) | $O(\sqrt{r}(\text{tr}(L_T^{1/4}) + m(\rho_{1:T}^L)^{1/4})(\text{tr}(R_T^{1/4}) + n(\rho_{1:T}^R)^{1/4}))$ | $O(\tau(m+n))$ | $O(\tau^2 mn)$ |
| **FTSL-Shampoo** (this work) | $O(\sqrt{r}(\text{tr}(L_T^{1/4}) + (\rho_{1:T}^L)^{1/4})(\text{tr}(R_T^{1/4}) + (\rho_{1:T}^R)^{1/4}))$ | $O(\tau(m+n))$ | $O(\tau^2 mn)$ |

## 2.1. OCO and Adaptive Subgradient Methods

Online convex optimization (OCO) is a powerful paradigm for solving sequential decision-making problems (Hazan, 2016; Orabona, 2019; Zhang et al., 2018; 2022). Specifically, it is typically formulated as an iterative game between a player and an adversary. In each round $t \in [T]$, the player begins by selecting a decision $\mathbf{x}_t \in \mathbb{R}^d$. After that, the adversary chooses a convex loss function $f_t(\cdot): \mathbb{R}^d \mapsto \mathbb{R}$, and the player incurs a loss $f_t(\mathbf{x}_t)$. The goal of the player is to minimize the cumulative loss $\sum_{t=1}^T f_t(\mathbf{x}_t)$ over $T$ rounds, which is equivalent to minimizing the regret (Zinkevich, 2003)

$$R(T) \triangleq \sum_{t=1}^T f_t(\mathbf{x}_t) - \sum_{t=1}^T f_t(\mathbf{x}^*), \quad (1)$$

defined as the excess loss suffered by the player compared to the loss of the fixed optimal choice $\mathbf{x}^* \in \arg\min_{\mathbf{x} \in \mathbb{R}^d} \sum_{t=1}^T f_t(\mathbf{x})$. Although Zinkevich (2003) establishes the optimal regret bound of $O(\sqrt{T})$, it is data-independent. In the following, we will introduce ADA-GRAD (Duchi et al., 2010a; 2011), a widely-used adaptive subgradient method, in both the primal-dual subgradient framework (Xiao, 2009) and the mirror descent framework (Duchi et al., 2010b), which achieves a data-dependent regret bound.

ADAGRAD can be categorized into two forms based on how the preconditioner $\tilde{G}_t$ is computed: ADAGRAD with full matrices (ADA-FULL) and ADAGRAD with diagonal matrices (ADA-DIAG). We denote $\mathbf{g}_t$ be a particular vector in the subdifferential set $\partial f_t(\mathbf{x}_t)$. Since we do not require the loss function to be smooth, we will not explicitly distinguish subgradients and gradients in the subsequent discussion. ADA-FULL first calculates the outer product matrix of the past gradients $G_t = \sum_{i=1}^t \mathbf{g}_i \mathbf{g}_i^\top$, and further

defines a symmetric matrix $\tilde{G}_t = \epsilon I_{d \times d} + G_t^{1/2}$, where $\epsilon > 0$ is a hyper-parameter introduced to ensure the invertibility of $\tilde{G}_t$. According to the primal-dual framework, the update rule is given by

$$\mathbf{x}_{t+1} = \arg\min_{\mathbf{x} \in \mathbb{R}^d} \left\{ \eta \left\langle \frac{1}{t}\overline{\mathbf{g}}_t, \mathbf{x} \right\rangle + \frac{1}{t}\Psi_t(\mathbf{x}) \right\}$$
$$= -\eta \tilde{G}_t^{-1} \overline{\mathbf{g}}_t,$$

where $\eta$ is the learning rate, $\overline{\mathbf{g}}_t = \sum_{i=1}^t \mathbf{g}_i$ is the sum of the received gradients and $\Psi_t(\mathbf{x}) = \frac{1}{2}\langle \mathbf{x}, \tilde{G}_t \mathbf{x} \rangle$ is the proximal term. The mirror descent version updates the decision as follows

$$\mathbf{x}_{t+1} = \arg\min_{\mathbf{x} \in \mathbb{R}^d} \left\{ \eta \langle \mathbf{g}_t, \mathbf{x} \rangle + B_{\Psi_t}(\mathbf{x}, \mathbf{x}_t) \right\}$$
$$= \mathbf{x}_t - \eta \tilde{G}_t^{-1} \mathbf{g}_t,$$

where $B_{\Psi_t}(\mathbf{x}, \mathbf{y}) = \Psi_t(\mathbf{x}) - \Psi_t(\mathbf{y}) - \langle \nabla \Psi_t(\mathbf{y}), \mathbf{x} - \mathbf{y} \rangle$ is the Bregman divergence associated with $\Psi_t(\cdot)$. ADA-FULL achieves an $O(\text{tr}(G_T^{1/2}))$ regret bound within the both frameworks. However, ADA-FULL store the past gradient outer products, i.e., $G_t$, and compute $G_t^{-1/2}$, resulting in $O(d^2)$ and $O(d^3)$ space and time complexities, respectively. The high computational cost of ADA-FULL renders it unsuitable for out-of-the-box use in scenarios with limited resources (Sagun et al., 2017; Ghorbani et al., 2019; Sankar et al., 2021; Zhou, 2024).

Different from ADA-FULL, ADA-DIAG only utilizes the diagonal elements of the gradient outer product matrix, i.e., redefining $\tilde{G}_t = \epsilon I_{d \times d} + \text{diag}(G_t)^{1/2}$, which thus is computationally more efficient. However, since the preconditioning matrix of ADA-DIAG only contains limited information, the regret bound of ADA-DIAG is worse than that of ADA-FULL when high-dimensional data is dense and has a low-rank structure.

## 2.2. Adaptive Subgradient Methods with Sketching

To alleviate the computational burden of ADA-FULL, several works have employed the sketching techniques to reduce its space and time complexities (Krummenacher et al., 2016; Wan & Zhang, 2018; Wan et al., 2018; Wan & Zhang, 2020; 2022; Feinberg et al., 2023). Krummenacher et al. (2016) propose ADA-LR to enhance the computational complexity of ADA-FULL by using random projection (Indyk & Motwani, 1998; Achlioptas, 2003). While ADA-LR reduces the time complexity to $O(\tau d^2)$, its space complexity remains at $O(d^2)$, where $\tau \ll d$ is the sketching size. To further improve the efficiency, they develop RADAGRA, which incorporates a more randomized approximation, achieving space and time complexities of $O(\tau d)$ and $O(\tau^2 d)$, respectively. However, RADAGRA is not supported by rigorous theoretical analysis. Later, Wan & Zhang (2018) develop ADA-DP based on random projection, which achieves space and time complexities of $O(\tau d)$ and $O(\tau^2 d)$, while providing theoretical guarantees.

Another class of sketching-based adaptive subgradient methods adopts frequent directions (FD) (Ghashami et al., 2016), a stable matrix sketching technique. Wan et al. (2018) first apply FD with ADA-FULL under both the primal-dual subgradient framework and the mirror descent framework by maintaining a matrix $B_t \in \mathbb{R}^{d \times \tau}$, such that $B_t B_t^\top \approx G_t \in \mathbb{R}^{d \times d}$, where $G_t$ represents the gradient covariance matrix. Their approach, ADA-FD, obtains space and time complexities of $O(\tau d)$ and $O(\tau^2 d)$, respectively. ADA-FD achieves regret bounds of $O(\text{tr}(G_T^{1/2}) + \sum_{t=1}^T \sqrt{\rho_t})$ and $O(\text{tr}(G_T^{1/2}) + \sum_{t=1}^T \tau\sqrt{\rho_t})$ under the primal-dual subgradient framework and the mirror descent framework, where $\rho_t$ is the removed eigenvalue of FD in round $t$. Furthermore, Wan & Zhang (2022) introduce a fast variant of ADA-FD, named ADA-FFD, by doubling the sketching size. ADA-FFD improves the time complexity to $O(\tau d)$ while keeping the same regret bounds. Although ADA-FD and ADA-FFD enjoy better space and time complexities, as mentioned by Feinberg et al. (2023), their regret bounds are $\Omega(T^{3/4})$ in some cases. For this reason, Feinberg et al. (2023) propose S-ADA by adding back the discarded information of FD to the FD-based preconditioner instead of utilizing a fixed diagonal regularization. While S-ADA enjoys an $O(\text{tr}(G_T^{1/2}) + \sqrt{d(d-\tau)\rho_{1:T}})$ regret bound, it suffers from a linear dependence on the dimensionality $d$. Moreover, it only achieves an unsatisfactory time complexity of $O(\tau^2 d)$.

Additionally, we also notice that FD has been utilized to accelerate online Newton step (ONS) algorithm (Hazan et al., 2007) for exponentially concave functions, and LinUCB (Chu et al., 2011) algorithm in linear contextual bandit setting, which also need to maintain a covariance matrix. Luo et al. (2016) first apply FD in ONS to construct a low-rank approximation of the matrix. To reduce the approximation error of FD, Luo et al. (2019) propose a new sketching strategy called robust frequent directions (RFD), which is the first method that compensates the discarded singular values back into the second-order matrix. They utilize RFD to propose a hyperparameter-free variant of ONS, which is more robust than FD-SON. In linear contextual bandit setting, Chen et al. (2020) propose spectral compensation frequent directions (SCFD) to approximate the covariance matrices, which adds up the total mass of subtracted values during FD procedure. SCFD can approximate a sequence of incremental covariance matrices while keeping the positive definite monotonicity. In fact, S-ADA can be viewed as a combination of ADA-FULL with SCFD.

## 2.3. Shampoo

Shampoo (Gupta et al., 2018) is an adaptive optimization method that takes the structure of the parameter space into consideration and thus is more efficient than ADA-FULL in scenarios where the decision is a matrix. Specifically, Shampoo maintains a set of preconditioning matrices, each of which operates on one dimension, while aggregating information across the remaining dimensions. For example, for a parameter matrix $X_t \in \mathbb{R}^{m \times n}$ and its gradient $G_t^X \in \mathbb{R}^{m \times n}$, ADA-FULL treats the matrix-shaped gradient as a vector of size $mn$ and its preconditioner $\tilde{G}_t$ has the size of $mn \times mn$, which leads to $O(m^2 n^2)$ and $O(m^3 n^3)$ space and time complexities, respectively. In contrast, Shampoo constructs two smaller matrices, $L_t \in \mathbb{R}^{m \times m}$ and $R_t \in \mathbb{R}^{n \times n}$, to precondition the rows and columns of $G_t^X$, respectively, which only requires an $O(m^2 + n^2)$ memory cost and an $O(m^3 + n^3)$ computation complexity. Since the parameters in the deep learning tasks often have matrix structures, Shampoo has strong empirical performance and receives lots of attentions (Anil et al., 2020; Liu et al., 2023; Eschenhagen et al., 2024).

However, the memory demands of Shampoo may still be prohibitive for large-scale neural networks. Anil et al. (2020) address the memory cost of Shampoo by introducing two variants. The first variant, Blocked Shampoo, partitions the decision variable $X_t \in \mathbb{R}^{m \times n}$ into $mn/b^2$ blocks, where $b$ is the block size and $b \le \min(m, n)$. However, Blocked Shampoo depends on the specific ordering of neurons in the hidden layers. The second variant relies on one-sided covariance upper bounds, which cannot effectively handle vector parameters. Feinberg et al. (2023) first incorporate FD into Shampoo and reduce its memory to $O(\tau(m+n))$. Their method, named Sketchy-Shampoo (S-Shampoo), uses two low-rank matrices $\hat{L}_t \in \mathbb{R}^{m \times \tau}$ and $\hat{R}_t \in \mathbb{R}^{n \times \tau}$ to approximate the preconditioning matrices $L_t$ and $R_t$, respectively. While S-Shampoo improves the space complexity of Shampoo, its regret bound relies on the dimensions $m, n$, leading to the unsatisfactory performance when the dimensions are high.

## 3. Preliminaries

### 3.1. Assumptions

We adopt two common assumptions of OCO (Hazan, 2016).

**Assumption 3.1.** All loss functions $f_t(\cdot)$ are convex.

**Assumption 3.2.** The optimal decision $\mathbf{x}^* \in \mathbb{R}^d$ is bounded by $D$, i.e., $\|\mathbf{x}^*\| \leq D$.

Besides, we introduce two assumptions for the scenario where the decision has a matrix structure. These assumptions have also been used in prior works (Gupta et al., 2018; Feinberg et al., 2023).

**Assumption 3.3.** The rank of gradient matrix $G_t^X$ is bounded by $r$, i.e., $\max_{t \in [T]} \operatorname{rank}(G_t^X) \leq r$.

**Assumption 3.4.** The optimal parameter $X^* \in \mathbb{R}^{m \times n}$ is bounded by $D_{\mathcal{M}}$, i.e., $\|X^*\|_F \leq D_{\mathcal{M}}$.

### 3.2. Frequent Directions

Frequent directions (FD) (Ghashami et al., 2016) is a deterministic matrix sketching technique by extending the well-known algorithm for approximating item frequencies in online data streams (Misra & Gries, 1982). For a given matrix $A \in \mathbb{R}^{d \times t}$, FD aims to generate a matrix $B \in \mathbb{R}^{d \times \tau}$ such that $BB^\top \approx AA^\top$, where $\tau \ll \min\{t, d\}$ is the sketching size. The procedure is summarized in Algorithm 1. In each round $t$, we denote the low-rank matrix $B_{t-1} = [\mathbf{b}_1, \mathbf{b}_2, ..., \mathbf{b}_{\tau-1}, \mathbf{0}_d] \in \mathbb{R}^{d \times \tau}$, where the last column is $\mathbf{0}_d$. Upon receiving the new gradient $\mathbf{g}_t \in \mathbb{R}^d$, it is inserted into the last column of $B_{t-1}$. Next, we perform singular value decomposition (SVD) on $B_{t-1} = U_t \sqrt{\operatorname{diag}(\lambda_{[1:\tau]}^{(t)})} V_t^\top$, and the matrix $B_t$ is computed as $B_t = U_t \sqrt{\operatorname{diag}(\lambda_{[1:\tau]}^{(t)} - \lambda_\tau^{(t)})}$ with its last column set to $\mathbf{0}_d$. The time complexity of FD is $O(\tau^2 d)$ for each iteration, which is dominated by computing the SVD of $B_{t-1}$, causing a quadratic dependence on sketching size $\tau$.

To further reduce the time complexity of FD, Ghashami et al. (2016) propose fast frequent directions (FFD) by expanding the space of $B_t$. Specifically, FFD maintains a matrix $B_0 = \mathbf{0}_{d \times 2\tau} \in \mathbb{R}^{d \times 2\tau}$. In each round $t$, we insert the received gradient $\mathbf{g}_t$ into the first all-zero column of $B_{t-1}$. Once $B_t$ no longer contains any all-zero columns, we perform SVD to obtain $B_t = U_t \sqrt{\operatorname{diag}(\lambda_{[1:2\tau]}^{(t)})} V_t^\top$. Then, the matrix $B_t$ is updated as $B_t = U_t \sqrt{\operatorname{diag}(\max\{\lambda_{[1:2\tau]}^{(t)} - \lambda_\tau^{(t)}, 0\})}$, ensuring that the last $\tau + 1$ columns are set to $\mathbf{0}_d$. As we only need to update the matrix $B_t$ every $\tau + 1$ rounds, the time complexity of FFD is $O(\tau d)$.

Since FD removes a singular value per round, the matrix $B_t B_t^\top$ does not preserve monotonicity. To resolve this limitation, Chen et al. (2020) propose spectral compensation

---

**Algorithm 1** Frequent Directions (FD)

---
1: **Input:** Sketching matrix $B_{t-1} \in \mathbb{R}^{d \times \tau}$ (with its last column as $\mathbf{0}_d$), new gradient vector $\mathbf{g}_t \in \mathbb{R}^d$
2: Insert the gradient $\mathbf{g}_t$ into the last column of $B_{t-1}$
3: Perform SVD to $B_{t-1} = U_t \sqrt{\operatorname{diag}(\lambda_{[1:\tau]}^{(t)})} V_t^\top$, where $U_t \in \mathbb{R}^{d \times \tau}$
4: Compute $B_t = U_t \sqrt{\operatorname{diag}(\lambda_{[1:\tau]}^{(t)} - \lambda_\tau^{(t)})}$
5: **Return:** $B_t$ and $\lambda_\tau^{(t)}$

---

frequent directions (SCFD), which adds up the total mass of subtracted values $\sum_{i=1}^t \lambda_\tau^{(t)}$ during FD procedure. SCFD is able to approximate a sequence of high-dimensional matrices while preserving positive definite monotonicity, i.e., $\sum_{i=1}^t \lambda_\tau^{(i)} I_{d \times d} + B_t B_t^\top \succeq \sum_{i=1}^{t-1} \lambda_\tau^{(i)} I_{d \times d} + B_{t-1} B_{t-1}^\top$.

## 4. The Proposed Methods

In this section, we first present FTSL, which incorporates FD with ADA-FULL under the primal-dual framework to obtain a better regret bound. Furthermore, we accelerate FTSL by employing an accelerated trick for FD, achieving enhanced computational efficiency. We demonstrate that this technique can be applied to expediting S-ADA (Feinberg et al., 2023). Additionally, we consider optimization problems involving matrix variables and propose an improved FD-based variant of Shampoo.

### 4.1. Our Improved Result

Before introducing our algorithms, we first briefly discuss why the regret bound of S-ADA (Feinberg et al., 2023) relies on the dimensionality $d$, offering motivation for the methods we design. Since S-ADA is under the mirror descent framework, its regret bound contains the Bregman divergence term, that is,

$$
O\left(\sum_{t=0}^{T-1} \left[ B_{\Psi_{t+1}}(\mathbf{x}^*, \mathbf{x}_{t+1}) - B_{\Psi_t}(\mathbf{x}^*, \mathbf{x}_{t+1}) \right]\right)
$$
$$
= O\left(\sum_{t=0}^{T-1} \|\mathbf{x}_{t+1} - \mathbf{x}_*\|_{\widetilde{G}_{t+1}^{1/2} - \widetilde{G}_t^{1/2}}^2\right),
$$

where $\Psi_t(\mathbf{x}) = \frac{1}{2}\langle \mathbf{x}, \tilde{G}_t^{1/2} \mathbf{x}\rangle$ is the proximal term, $\tilde{G}_t$ is the preconditioning matrix and $B_{\Psi_t}(\mathbf{x}, \mathbf{y}) = \Psi_t(\mathbf{x}) - \Psi_t(\mathbf{y}) - \langle \nabla \Psi_t(\mathbf{y}), \mathbf{x} - \mathbf{y}\rangle$. To facilitate summation, they upper bound this term by $O(\sum_{t=0}^{T-1} \operatorname{tr}(\tilde{G}_{t+1}^{1/2} - \tilde{G}_t^{1/2}))$ and then exploit the additivity of the trace, which yields a bound of $O(\operatorname{tr}(\tilde{G}_T^{1/2}))$. Feinberg et al. (2023) add the cumulative removed eigenvalues of FD $\rho_{1:t}$ into $\tilde{G}_t$. Consequently, the Bregman divergence term is $O(\operatorname{tr}((B_T B_T^\top + \rho_{1:T} I_{d \times d})^{1/2}))$, which is further bounded by $O(\operatorname{tr}(G_T^{1/2}) +$

---

**Algorithm 2** Follow the Sketchy Leader (FTSL)

1: **Input:** Learning rate $\eta$, sketching size $\tau$
2: Initialize $\mathbf{x}_1 = \mathbf{0}_d, \overline{\mathbf{g}}_0 = \mathbf{0}_d, \tilde{G}_0 = \mathbf{0}_{d \times d}, B_0 = \mathbf{0}_{d \times \tau}$
3: **for** $t = 1$ to $T$ **do**
4:     Play the decision $\mathbf{x}_t$ and suffer the loss $f_t(\mathbf{x}_t)$
5:     Query the gradient $\mathbf{g}_t = \nabla f_t(\mathbf{x}_t)$ and calculate $\overline{\mathbf{g}}_t = \overline{\mathbf{g}}_{t-1} + \mathbf{g}_t$
6:     Send $B_{t-1}$ and $\mathbf{g}_t$ to Algorithm 1
7:     Receive $B_t$ and set $\rho_t = \lambda_\tau^t$
8:     Calculate $\tilde{G}_t = B_t B_t^\top + \rho_{1:t} I_{d \times d}$ and derive $\tilde{G}_t^{-1/2}$
9:     Update $\mathbf{x}_{t+1}$ according to (2)
10: **end for**

---

$\sqrt{d(d-\tau)\rho_{1:T}}$), where $B_T \in \mathbb{R}^{d \times \tau}$ is the sketching matrix and $G_T = \sum_{t=1}^T \mathbf{g}_t \mathbf{g}_t^\top$. As a result, the regret bound of S-ADA exhibits a *linear* dependence on $d$, resulting in an unsatisfactory performance in high-dimensional problems.

To overcome this issue, we propose integrating FD with ADA-FULL under *the primal-dual subgradient framework*. Our method, which we call FTSL, is outlined in Algorithm 2. Specifically, we employ the FD to construct a low-rank approximation of the outer product matrix of gradients $G_t$, aiming to reduce the computational complexity. To ensure the monotonicity of the preconditioning matrix $\tilde{G}_t$, we also add back the cumulative escaped masses $\rho_{1:t}$ into $\tilde{G}_t$. Under the primal-dual subgradient framework, we update the decision as follows

$$
\begin{aligned}
\mathbf{x}_{t+1} &= \arg\min_{\mathbf{x} \in \mathbb{R}^d} \{\eta \langle \overline{\mathbf{g}}_t, \mathbf{x} \rangle + \Psi_t(\mathbf{x})\} \\
&= -\eta \tilde{G}_t^{-1/2} \overline{\mathbf{g}}_t,
\end{aligned}
\tag{2}
$$

where $\overline{\mathbf{g}}_t = \sum_{i=1}^t \mathbf{g}_i$ is the sum of the past gradients. According to the analysis under the primal-dual framework, the regret of FTSL is upper bounded by the term $O(\|\tilde{G}_T^{1/2}\|) = O(\|(B_T B_T^\top + \rho_{1:T} I_{d \times d})^{1/2}\|) \leq O(\|G_T^{1/2}\| + \sqrt{\rho_{1:T}})$, thereby avoiding the dependence on $d$.

Formally, we present the theoretical guarantee of FTSL.

**Theorem 4.1.** *Under Assumption 3.1 and Assumption 3.2, by setting the learning rate* $\eta = \frac{D}{\sqrt{2}}$, *FTSL ensures*

$$
R(T) \leq O\left(\mathrm{tr}(G_T^{1/2}) + \sqrt{\rho_{1:T}}\right),
$$

*where* $G_T = \sum_{t=1}^T \mathbf{g}_t \mathbf{g}_t^\top$.

**Remark.** In contrast to the previous regret bound of $O(\mathrm{tr}(G_T^{1/2}) + \sqrt{d(d-\tau)\rho_{1:T}})$ (Feinberg et al., 2023), the regret bound of FTSL is *dimension-free*, a benefit realized from the primal-dual subgradient framework.

**Remark.** Since we only maintain a sketching matrix $B_t \in \mathbb{R}^{d \times \tau}$, the space complexity of FTSL is $O(\tau d)$. Its time

---

**Algorithm 3** Follow the Fast Sketchy Leader (FTFSL)

1: **Input:** Learning rate $\eta$, sketching size $\tau$
2: Initialize $\mathbf{x}_1 = \mathbf{0}_d, \tilde{G}_0 = \mathbf{0}_{d \times d}, r_0 = 0, M_0 = \mathbf{0}_{2\tau \times 2\tau}, V_0 = \mathbf{0}_{d \times 2\tau}, \overline{\mathbf{g}}_0 = \mathbf{0}_d, \rho_1 = 0$
3: **for** $t = 1$ to $T$ **do**
4:     Play the decision $\mathbf{x}_t$ and suffer the loss $f_t(\mathbf{x}_t)$
5:     Query the gradient $\mathbf{g}_t = \nabla f_t(\mathbf{x}_t)$ and compute $\mathbf{g}'_t = V_{t-1}(V_{t-1}^\top \mathbf{g}_t), \overline{\mathbf{g}}_t = \overline{\mathbf{g}}_{t-1} + \mathbf{g}_t$
6:     **if** $\mathbf{g}'_t \neq \mathbf{g}_t$ **then**
7:         Set $r_{t-1} = r_{t-1} + 1$, calculate $\mathbf{v}_{r_{t-1}}^{t-1} = \frac{\mathbf{g}_t - \mathbf{g}'_t}{\|\mathbf{g}_t - \mathbf{g}'_t\|}$ and set the $r_{t-1}$-th column of $V_{t-1}$ as $\mathbf{v}_{r_{t-1}}^{t-1}$
8:     **end if**
9:     Set $r_t = r_{t-1}, V_t = V_{t-1}$
10:     Compute $M_t = M_{t-1} + (V_{t-1}^\top \mathbf{g}_t)(V_{t-1}^\top \mathbf{g}_t)^\top$
11:     Perform SVD decomposition on $M_t$, which is $U_t \Sigma_t U_t^\top = U_t \mathrm{diag}(\lambda_{[1:2\tau]}^{(t)}) U_t^\top = M_t$
12:     Calculate $\tilde{G}_t = \rho_{1:t} I_{d \times d} + V_t U_t \Sigma_t U_t V_t^\top$
13:     Update $\mathbf{x}_{t+1}$ according to (2) and set $\rho_{t+1} = 0$
14:     **if** $r_t = 2\tau$ **then**
15:         Set $\rho_{t+1} = \lambda_\tau^{(t)}, M_t = \mathrm{diag}(\max\{\lambda_{[1:2\tau]}^{(t)} - \lambda_\tau^{(t)}, 0\})$ and $V_t = V_t U_t$
16:         Set $r_t = \tau - 1$ and the $\tau$-th to $2\tau$-th columns of $V_t$ be $\mathbf{0}_d$
17:     **end if**
18: **end for**

---

complexity is $O(\tau^2 d)$ per round, which arises from the SVD of $B_t$ and the calculation of $\tilde{G}_t^{-1/2}$ (the detailed discussions can be found in Appendix A). While the time complexity of FTSL is linear with respect to the dimensionality $d$, it still suffers from the quadratic dependence on the sketching size $\tau$. To further alleviate its computational burden, we develop a fast variant of FTSL in the next section.

### 4.2. Our Accelerated Variant

The time complexity of FTSL suffers from a quadratic dependence on the sketching size $\tau$, which is introduced by the SVD decomposition on $B_t \in \mathbb{R}^{d \times \tau}$ in FD. Drawing inspiration from the previous work (Chen et al., 2020; Wan & Zhang, 2022), we adopt a more efficient strategy for computing the SVD of sketching matrix $B_t$. Our method, termed FTFSL, is presented in Algorithm 3.

Different from FD, the sketching matrix $B_t$ is expanded to $\mathbb{R}^{d \times 2\tau}$ in FFD. Rather than explicitly maintaining $B_t$, we use two matrices $V_t$ and $M_t$ to form $B_t$. Specifically, $V_t = [\mathbf{v}_1^t, \cdots, \mathbf{v}_{2\tau}^t] \in \mathbb{R}^{d \times 2\tau}$ consists of $r_t$ orthonormal vectors ($r_t \leq 2\tau$) and the rest columns are zero vectors, and $M_t \in \mathbb{R}^{2\tau \times 2\tau}$ is a symmetric matrix. We require that $V_t$ and $M_t$ satisfy the condition $V_t M_t^{1/2} = B_t \in \mathbb{R}^{d \times 2\tau}$. In each round $t$, after receiving the gradient $\mathbf{g}_t$, we first check whether this

vector lies within the subspace spanned by $V_{t-1}$. If the vector is not contained within the subspace, we normalize it and subsequently add it to $V_{t-1}$, thereby enlarging the span of the subspace and ensuring $V_{t-1}V_{t-1}^\top \mathbf{g}_t = \mathbf{g}_t$ (Step 6-8). Then we have the following equation

$$V_{t-1}M_{t-1}V_{t-1}^\top + \mathbf{g}_t\mathbf{g}_t^\top$$
$$= V_{t-1}\left(M_{t-1} + V_{t-1}^\top \mathbf{g}_t\mathbf{g}_t^\top V_{t-1}\right)V_{t-1}^\top.$$

This implies that we only need to perform an SVD decomposition on $M_{t-1} + (V_{t-1}^\top \mathbf{g}_t)(V_{t-1}^\top \mathbf{g}_t)^\top \in \mathbb{R}^{2\tau \times 2\tau}$, which only takes a time complexity of $O(\tau^3)$. Next, we incorporate the escaped masses to keep the monotonicity of the preconditioning matrix $\tilde{G}_t$. When $r_t = 2\tau$, we need to set $r_t = \tau - 1$ and set the last $\tau + 1$ columns of the sketching matrix $B_t$ to be zero (Step 14-16). Given the decomposition $M_t = U_t \operatorname{diag}(\lambda_{[1:2\tau]}^{(t)})U_t^\top$ and the relationship $V_t M_t^{1/2} = B_t$, this can be efficiently achieved by updating $M_t$ as $\operatorname{diag}(\max\{\lambda_{[1:2\tau]}^{(t)} - \lambda_\tau^{(t)}, 0\})$, calculating $V_t = V_t U_t$, and setting the last $\tau + 1$ columns of $V_t$ to zero.

In the following, we provide the theoretical guarantee of FTFSL.

**Theorem 4.2.** *Under Assumption 3.1 and Assumption 3.2, by setting the learning rate $\eta = \frac{D}{\sqrt{2}}$, FTFSL ensures*

$$R(T) \le O\left(\operatorname{tr}(G_T^{1/2}) + \sqrt{\rho_{1:T}}\right).$$

**Remark.** In each round, FTFSL computes $\mathbf{g}'_t, \mathbf{v}_{r_t-1}^{t-1}, M_t$, SVD of $M_t$ and update $\mathbf{x}_t$, with respective time complexities of $O(\tau d)$, $O(d)$, $O(\tau d)$, $O(\tau^3)$ and $O(\tau d)$. Additionally, we only compute $V_t = V_t U_t$ every $\tau + 1$ rounds, incurring a time complexity of $O(\tau^2 d)$. When $\tau \le O(\sqrt{d})$, the time complexity of FTSL is $O(\tau d)$ per round.

Notably, we can also reduce the time complexity of S-ADA (Feinberg et al., 2023) by adopting this technique. We replace the update rule for the decision variable $\mathbf{x}_t$ of FTFSL (Step 13) with the following

$$\mathbf{x}_{t+1} = \mathbf{x}_t - \eta \tilde{G}_t^{-1/2}\mathbf{g}_t,$$

and analyze under the mirror descent framework.

Then we provide the theoretical guarantee of Fast S-ADA.

**Theorem 4.3.** *Under Assumption 3.1 and assuming $D_1 = \max_{t\in[T]}\|\mathbf{x}_t - \mathbf{x}^*\|$, by setting the learning rate $\eta = \frac{D_1}{\sqrt{2}}$, Fast S-ADA ensures*

$$R(T) \le O\left(\operatorname{tr}(G_T^{1/2}) + \sqrt{d(d-\tau)\rho_{1:T}}\right).$$

**Remark.** Compared to S-ADA (Feinberg et al., 2023), Fast S-ADA obtains a better $O(\tau d)$ time complexity when $\tau \le O(\sqrt{d})$, while preserving the same regret bound.

---

**Algorithm 4** Frequent Directions in General Form

1: **Input:** Sketching matrix $B_{t-1} \in \mathbb{R}^{d\times\tau}$, a new symmetric PSD matrix $M_t \in \mathbb{R}^{d\times d}$
2: Eigendecompose $\overline{U}_t\operatorname{diag}(\lambda^{(t)})\overline{U}_t^\top = B_{t-1}B_{t-1}^\top + M_t$, define $U_t \in \mathbb{R}^{d\times\tau}$ be the first $\tau$ columns of $\overline{U}_t$ and $\lambda_{[1:\tau]}^{(t)}$ be its eigenvalues
3: Compute $B_t = U_t\sqrt{\operatorname{diag}(\lambda_{[1:\tau]}^{(t)} - \lambda_\tau^{(t)})}$
4: **Return:** $B_t$ and $\lambda_\tau^{(t)}$

---

**Algorithm 5** FTSL-Shampoo

**Require:** Learning rate $\eta$, sketching size $\tau$, $\epsilon > 0$
1: Initialize $X_1 = \mathbf{0}_{m\times n}, \hat{L}_0 = \mathbf{0}_{m\times\tau}, \hat{R}_0 = \mathbf{0}_{n\times\tau}, \tilde{L}_0 = \mathbf{0}_{m\times m}, \tilde{R}_0 = \mathbf{0}_{n\times n}, \overline{G}_0^X = \mathbf{0}_{m\times n}$
2: **for** $t = 1$ to $T$ **do**
3:    Play the decision $X_t$ and suffer the loss $f_t(X_t)$
4:    Query the gradient $G_t^X = \nabla f_t(X_t) \in \mathbb{R}^{m\times n}$ and calculate $\overline{G}_t^X = \overline{G}_{t-1}^X + G_t^X$
5:    Send $\hat{L}_{t-1}$ and $G_t^X(G_t^X)^\top$ to Algorithm 4 and receive $\hat{L}_t, \rho_t^L$
6:    Send $\hat{R}_{t-1}$ and $(G_t^X)^\top G_t^X$ to Algorithm 4 and receive $\hat{R}_t, \rho_t^R$
7:    Update $\tilde{L}_t = \hat{L}_t\hat{L}_t^\top + (\epsilon + \rho_{1:t}^L)I_{m\times m}$
8:    Update $\tilde{R}_t = \hat{R}_t\hat{R}_t^\top + (\epsilon + \rho_{1:t}^R)I_{n\times n}$
9:    Update $X_{t+1}$ according to (3)
10: **end for**

---

### 4.3. Optimization Problems with Matrix Variables

In this section, we consider a practical scenario where the decision variable is a matrix $X_t \in \mathbb{R}^{m\times n}$, which is common for parameters in deep learning tasks. In such settings, the loss $f(X)$ is typically a smooth non-convex function, and the objective is to find a point $X_T$ such that $\|\nabla f(X_T)\| \le \epsilon$. As pointed out by Agarwal et al. (2019), a smooth non-convex problem can be transformed into solving a series of offline convex problems by using the online to batch conversion. Therefore, we can derive the non-convex optimization guarantees from online regret bounds, with further details provided in Appendix B.

To utilize the structure information, Gupta et al. (2018) propose Shampoo, which retains the matrix structure of the gradient and maintains two matrices as preconditioners of the rows and columns of $G_t^X$, yielding a space complexity of $O(m^2 + n^2)$. While S-Shampoo (Feinberg et al., 2023) improves the space complexity of Shampoo to $O(\tau(m+n))$, its regret bound again relies on the dimensionality $m, n$. To further reduce its regret bound, we propose FTSL-Shampoo by integrating FD with Shampoo under the primal-dual framework. Our method achieves a superior dimension-free guarantee with obtaining an $O(\tau(m+n))$ space complexity,

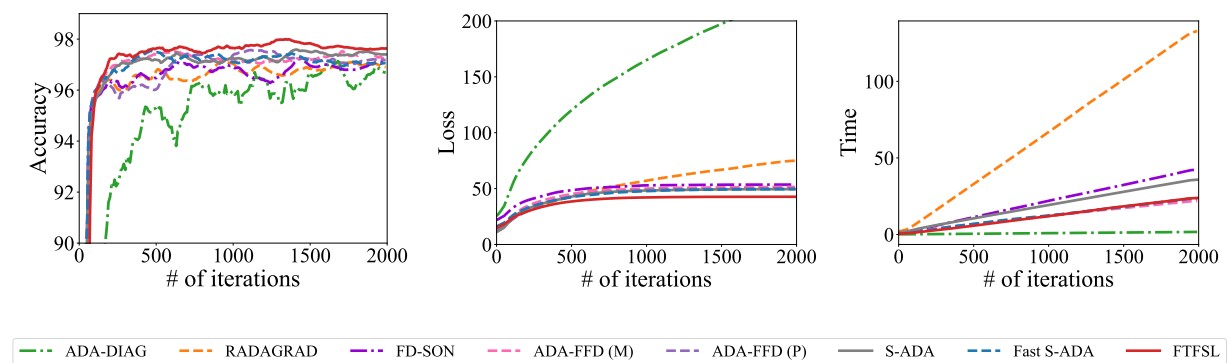

*Figure 1.* Results for Gisette dataset.

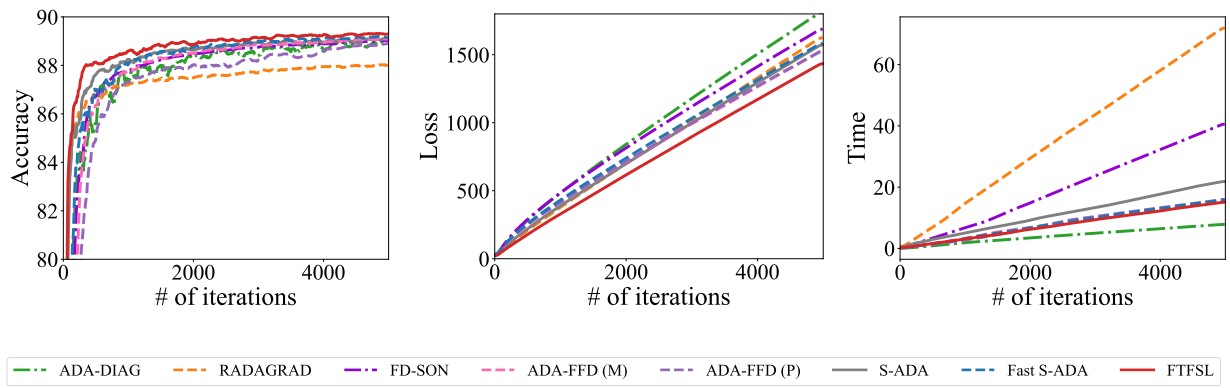

*Figure 2.* Results for Epsilon dataset.

which is presented in Algorithm 5.

Specifically, we utilize FD to approximate the left and right preconditioning matrices for Shampoo. We maintain two matrices $\hat{L}_t \in \mathbb{R}^{m \times \tau}, \hat{R}_t \in \mathbb{R}^{n \times \tau}$ to ensure that $\hat{L}_t \hat{L}_t^\top \approx \sum_{i=1}^t G_i^X (G_i^X)^\top, \hat{R}_t \hat{R}_t^\top \approx \sum_{i=1}^t (G_i^X)^\top G_i^X$. We track the cumulative escaped masses $\rho_{1:t}^L$ and $\rho_{1:t}^R$ of the left and right preconditioning matrices separately, and then add them back into $\tilde{L}_t$ and $\tilde{R}_t$ to uphold the monotonicity. To achieve a dimension-free regret bound with FD, we update the parameters as follows:

$$X_{t+1} = -\eta \tilde{L}_t^{-1/4} \overline{G}_t^X \tilde{R}_t^{-1/4}, \tag{3}$$

where $\overline{G}_t^X = \sum_{i=1}^t G_i^X$ is the sum of the past gradients, and conduct the analysis under *the primal-dual framework*. Then we present the regret bound of FTSL-Shampoo.

**Theorem 4.4.** *Under Assumption 3.1, Assumption 3.3 and Assumption 3.4, by setting the learning rate $\eta = \frac{D_{\mathcal{M}}}{\sqrt{r}}$ and further denoting $L_T = \epsilon I_{m \times m} + \sum_{t=1}^T G_t^X (G_t^X)^\top, R_T =$*

$\epsilon I_{n \times n} + \sum_{t=1}^T (G_t^X)^\top G_t^X$, *FTSL-Shampoo ensures*

$$R(T) \leq O \left( \sqrt{r} (\text{tr}(L_T^{1/4}) + (\rho_{1:T}^L)^{1/4}) \right.$$
$$\left. \cdot (\text{tr}(R_T^{1/4}) + (\rho_{1:T}^R)^{1/4}) \right),$$

*where $\rho_{1:T}^L$ and $\rho_{1:T}^R$ represent the sum of the removed eigenvalues of FD during the approximation of $\sum_{t=1}^T G_t^X (G_t^X)^\top$ and $\sum_{t=1}^T (G_t^X)^\top G_t^X$, respectively.*

**Remark.** In comparison to the previous $O(\sqrt{r}(\text{tr}(L_T^{1/4}) + m(\rho_{1:T}^L)^{1/4})(\text{tr}(R_T^{1/4}) + n(\rho_{1:T}^R)^{1/4}))$ regret bound of S-Shampoo (Feinberg et al., 2023), we achieve a dimension-free regret bound while enjoying the same $O(\tau(m + n))$ space complexity.

## 5. Experiments

In this section, we assess the performance of the proposed methods via numerical experiments on online classification and image classification tasks. Due to the limited space, we only present a subset of the experimental outcomes, with the comprehensive set of results accessible in Appendix C.

**Online Classification.** First, we perform online classification to evaluate the performance of our methods with two real-world datasets from LIBSVM (Chang & Lin, 2011) repository: Gisette and Epsilon, which are high-dimensional and dense. Particularly, Gisette dataset contains 6000 training samples and 1000 testing samples, each with 5000 features. Epsilon dataset consists of $400,000$ training samples and $100,000$ testing samples, each with 2000 features. In each round $t \in [T]$, a batch of training examples $\{(\mathbf{w}_{t,1}, y_{t,1}), \ldots, (\mathbf{w}_{t,n}, y_{t,n})\}$ arrive, where $(\mathbf{w}_{t,i}, y_{t,i}) \in [-1,1]^d \times \{-1,1\}, i = 1, \ldots, n$. The online learner aims to predict a linear model $\mathbf{x}_t$ and suffers the hinge loss $f_t(\mathbf{x}_t) = \frac{1}{n} \sum_{i=1}^n \max\{0, 1 - y_t \mathbf{x}_t^\top \mathbf{w}_{t,i}\}$. For Gisette dataset, we set the batch size $n = 32$, the sketching size $\tau = 50$ to be $1\%$ of the original dimensionality, and $T = 2000$. For Epsilon dataset, we set the batch size $n = 128$, $\tau = 20$ and $T = 5000$.

**Results.** Following Duchi et al. (2011), we adopt the performance of accuracy on the testing data to compare different methods. To better demonstrate the improvements of our methods, we additionally plot the training loss and runtime of various methods. From Figure 1 and Figure 2, we observe that FTFSL outperforms all other methods in both loss and testing accuracy, aligning with its superior regret bound. Moreover, FTFSL and Fast S-ADA exhibit significantly lower runtimes compared to S-ADA, owing to their superior time complexities.

## 6. Conclusion

In this paper, we investigate adaptive subgradient methods with Frequent Directions (FD). First, we introduce a novel method, named FTSL, to achieve a tighter dimension-free regret bound of $O(\operatorname{tr}(G_T^{1/2}) + \sqrt{\rho_{1:T}})$. Next, we propose a fast version of FTSL by accelerating FD used in it, which improves the time complexity to $O(\tau d)$ while preserving the same regret bound. This technique can also be applied to expedite S-ADA (Feinberg et al., 2023). Additionally, we consider a more complex scenario where the decision is a matrix, and adapt FD to Shampoo under the primal-dual framework to obtain a better dimension-free bound. Finally, we substantiate the effectiveness and efficiency of our methods through experimental validation.

## Acknowledge

This work was partially supported by National Science and Technology Major Project (2022ZD0114801), NSFC (U23A20382), and Yongjiang Talent Introduction Programme (2023A-193-G). The authors would like to thank the anonymous reviewers for their constructive suggestions.

## Impact Statement

This paper presents work whose goal is to advance the field of Machine Learning. There are many potential societal consequences of our work, none which we feel must be specifically highlighted here.

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

## A. Proof of Theorems

**Notations.** We denote $\mathbf{x}$ to represent a vector and $X$ to represent a matrix. For any vector $\mathbf{x} \in \mathbb{R}^d$ and a positive semi-definite matrix $A \in \mathbb{R}^{d \times d}$, $\|\mathbf{x}\|_A^2 = \langle \mathbf{x}, A\mathbf{x} \rangle$. For a matrix $A$, $A^{-1}$ is the inverse of $A$ if $A$ is full rank; otherwise, $A^{-1}$ is taken to be the Moore-Penrose pseudoinverse. For two matrices $A, B$, $A \preceq B$ if and only if $B - A$ is a positive semi-definite matrix. we define the space complexity and time complexity as the memory and time usage in each round, respectively. For example, the time complexity of performing SVD to a matrix $A \in \mathbb{R}^{m \times n}$ is $O(\min\{m^2 n, n^2 m\})$. For simplicity, we use $\|\cdot\|$ for $\|\cdot\|_2$ by default. $\lambda_{[1:\tau]}$ is a sequence containing $\tau$ elements, and $\max\{\lambda_{[1:\tau]}, a\}$ represents a new sequence, where each element is the maximum of $\lambda_i$ and $a$, and $\mathrm{diag}(\lambda_{[1:\tau]})$ is a diagonal matrix where the $i$-th diagonal element is $\lambda_i$, $1 \leq i \leq 2\tau$. In the proof, we do not require smoothness of loss functions and we do not explicitly distinguish subgradients and gradients.

### A.1. Calculation of $\tilde{G}_t^{-1/2}$

In FTSL, we need to calculate $\tilde{G}_t^{-1/2}$, which can not be directly derived by Woodbury formula (Hager, 1989). We provide the following process.

Assume $U_t^\perp$ is the complementary subspace of $U_t \in \mathbb{R}^{d \times \tau}$, we have $I_{d \times d} = U_t U_t^\top + U_t^\perp (U_t^\perp)^\top$. We denote $\mathrm{diag}(\lambda_{[1:\tau]}^{(t)} - \lambda_\tau^{(t)}) = \Sigma_t \in \mathbb{R}^{\tau \times \tau}$ and have

$$\rho_{1:t} I_{d \times d} + U_t \Sigma_t U_t^\top = U_t(\Sigma_t + \rho_{1:t} I_{d \times d}) U_t^\top + \rho_{1:t} I_{d \times d} U_t^\perp (U_t^\perp)^\top$$
$$U_t(\Sigma_t + \rho_{1:t}) U_t^\top = \rho_{1:t} I_{d \times d} + U_t \Sigma_t U_t^\top - \rho_{1:t} U_t^\perp (U_t^\perp)^\top.$$

Then we can get

$$\rho_{1:t} I_{d \times d} + U_t \Sigma_t U_t^\top = U_t(\Sigma_t + \rho_{1:t}) U_t^\top + \rho_{1:t} U_t^\perp (U_t^\perp)^\top = [U_t; U_t^\perp] \Sigma_t' [U_t; U_t^\perp]^\top,$$

where $\Sigma_t' = \begin{bmatrix} \Sigma_t + \rho_{1:t} I_{\tau \times \tau} & \mathbf{0} \\ \mathbf{0} & \rho_{1:t} I_{d-\tau, d-\tau} \end{bmatrix}$.

Therefore, we have

$$\sqrt{\rho_{1:t} I_{d \times d} + U_t \Sigma_t U_t^\top} = U_t \sqrt{(\Sigma_t + \rho_{1:t} I_{\tau \times \tau})} U_t^\top + \sqrt{\rho_{1:t}} U_t^\perp (U_t^\perp)^\top$$
$$= \sqrt{\rho_{1:t}} I_{d \times d} + U_t(\sqrt{\Sigma_t + \rho_{1:t} I_{\tau \times \tau}} - \sqrt{\rho_{1:t} I_{\tau \times \tau}}) U_t^\top.$$

We can apply Woodbury formula (Hager, 1989) on it.

We can derive

$$(\rho_{1:t} I_{d \times d} + U_t \Sigma_t U_t^\top)^{-1/2} = \frac{1}{\sqrt{\rho_{1:t}}} \left( I_{d \times d} - U_t \left( \sqrt{\Sigma_t + \rho_{1:t} I_{\tau \times \tau}} \right)^{-1} \left( \sqrt{\Sigma_t + \rho_{1:t} I_{\tau \times \tau}} - \sqrt{\rho_{1:t} I_{\tau \times \tau}} \right) U_t^\top \right).$$

### A.2. Proof of Theorem 4.1

Before giving the proof of Theorem 4.1, we first introduce some supporting lemmas.

**Lemma A.1.** *(Proposition 2 in Duchi et al. (2011)) Let $\{\mathbf{x}_t\}$ be the decisions of Algorithm 2 and $\mathbf{x}^* \in \arg\min_{\mathbf{x} \in \mathbb{R}^d} \sum_{t=1}^T f_t(\mathbf{x})$, we have*

$$\sum_{t=1}^T f_t(\mathbf{x}_t) - \sum_{t=1}^T f_t(\mathbf{x}^*) \leq \frac{1}{\eta} \Psi_T(\mathbf{x}^*) + \frac{\eta}{2} \sum_{t=1}^T \|\mathbf{g}_t\|_{\tilde{G}_{t-1}^{-1/2}}^2,$$

*where $\Psi_T(\mathbf{x}^*) = \frac{1}{2} \left\langle \mathbf{x}^*, \tilde{G}_T^{1/2} \mathbf{x}^* \right\rangle$.*

**Lemma A.2.** *(Lemma 10 in Duchi et al. (2011)) Let $G_t = \sum_{i=1}^t \mathbf{g}_i \mathbf{g}_i^\top$. We have*

$$\sum_{t=1}^T \left\langle \mathbf{g}_t, \left(G_t^{1/2}\right)^{-1} \mathbf{g}_t \right\rangle \leq 2 \sum_{t=1}^T \left\langle \mathbf{g}_t, \left(G_T^{1/2}\right)^{-1} \mathbf{g}_t \right\rangle = 2\operatorname{tr}\left(G_T^{1/2}\right).$$

Then, we illustrate how FD approximates the original matrix.

**Lemma A.3.** *(Remark 11 in Feinberg et al. (2023)) With $G_t = \sum_{i=1}^t \mathbf{g}_i \mathbf{g}_i^\top$, for FD results in FTSL, we have*

$$B_t B_t^\top \preceq G_t \preceq \tilde{G}_t = \rho_{1:t} I_{d \times d} + B_t B_t^\top.$$

By using Lemma A.1, we can bound the regret of Algorithm 2 by

$$\sum_{t=1}^T f_t(\mathbf{x}_t) - \sum_{t=1}^T f_t(\mathbf{x}^*) \leq \underbrace{\frac{1}{\eta} \Psi_T(\mathbf{x}^*)}_{R_D} + \underbrace{\frac{\eta}{2} \sum_{t=1}^T \|\mathbf{g}_t\|_{\tilde{G}_{t-1}^{-1/2}}^2}_{R_G}. \tag{4}$$

Then we bound the term $R_D$ and $R_G$, respectively.

As for $R_D$, we have

$$\begin{aligned}
\Psi_T(\mathbf{x}^*) &= \frac{1}{2} \left\langle \mathbf{x}^*, \tilde{G}_T^{1/2} \mathbf{x}^* \right\rangle = \frac{1}{2} \left\langle \mathbf{x}^*, \left(B_T B_T^\top + \rho_{1:T} I\right)^{1/2} \mathbf{x}^* \right\rangle \\
&\leq \frac{1}{2} \left\langle \mathbf{x}^*, \left(B_T B_T^\top\right)^{1/2} \mathbf{x}^* \right\rangle + \frac{1}{2} \left\langle \mathbf{x}^*, \left(\rho_{1:T} I\right)^{1/2} \mathbf{x}^* \right\rangle \\
&\leq \frac{1}{2} \sqrt{\rho_{1:T}} \|\mathbf{x}^*\|^2 + \frac{1}{2} \left\langle \mathbf{x}^*, \left(G_T\right)^{1/2} \mathbf{x}^* \right\rangle \\
&\leq \frac{1}{2} D^2 \sqrt{\rho_{1:T}} + \frac{1}{2} D^2 \lambda_{\max}\left(\left(G_T\right)^{1/2}\right) \\
&\leq \frac{1}{2} D^2 \sqrt{\rho_{1:T}} + \frac{1}{2} D^2 \operatorname{tr}(G_T^{1/2}),
\end{aligned}$$

where the first inequality is due to $\sqrt{a+b} \leq \sqrt{a} + \sqrt{b}$ and we assume $\|\mathbf{x}^*\| \leq D$.

Therefore, we can get

$$R_D = \frac{1}{\eta} \Psi_T(\mathbf{x}^*) \leq \frac{1}{2\eta} \left(D^2 \sqrt{\rho_{1:T}} + D^2 \operatorname{tr}(G_T^{1/2})\right). \tag{5}$$

To give the bound for $R_G$, we first derive the lower bound of $\tilde{G}_{t-1}$, which connects $\tilde{G}_{t-1}$ with $G_t$. We denote $a_t = \max\{\frac{\rho_{1:t-1}}{\|\mathbf{g}_t\|^2 + \rho_{1:t-1}}, 1\} \leq 1$, and we have

$$\begin{aligned}
\tilde{G}_{t-1} &= B_{t-1} B_{t-1}^\top + \rho_{1:t-1} I_{d \times d} \\
&\succeq a_t(\|\mathbf{g}_t\|^2 I_{d \times d} + \rho_{1:t-1} I_{d \times d} + B_{t-1} B_{t-1}^\top) \\
&\succeq a_t(\|\mathbf{g}_t\|^2 I_{d \times d} + G_{t-1}) \\
&\succeq a_t G_t,
\end{aligned}$$

which means $\tilde{G}_{t-1}^{-1} \preceq \frac{1}{a_t} G_t^{-1}$.

Then, letting $C_1 = \max_{t \in [T]} \frac{1}{\sqrt{a_t}}$, we have

$$\begin{aligned}
\sum_{t=1}^T \|\mathbf{g}_t\|_{\tilde{G}_{t-1}^{-1/2}}^2 &= \sum_{t=1}^T \left\langle \mathbf{g}_t, (\tilde{G}_{t-1}^{1/2})^{-1} \mathbf{g}_t \right\rangle \leq \sum_{t=1}^T \frac{1}{\sqrt{a_t}} \left\langle \mathbf{g}_t, (G_t^{1/2})^{-1} \mathbf{g}_t \right\rangle \\
&\leq C_1 \sum_{t=1}^T \left\langle \mathbf{g}_t, (G_t^{1/2})^{-1} \mathbf{g}_t \right\rangle \leq 2C_1 \operatorname{tr}(G_T^{1/2}),
\end{aligned}$$

where the last inequality is due to Lemma A.2.

Therefore, we can get

$$R_G = \frac{\eta}{2} \sum_{t=1}^{T} \|\mathbf{g}_t\|_{\tilde{G}_{t-1}^{-1/2}}^2 \leq \eta C_1 \operatorname{tr}(G_T^{1/2}). \tag{6}$$

By combining inequalities (5) and (6), and setting $\eta = \frac{D}{\sqrt{2}}$, we obtain the regret bound for FTSL

$$R(T) = \sum_{t=1}^{T} f_t(\mathbf{x}_t) - \sum_{t=1}^{T} f_t(\mathbf{x}^*) \leq \eta C_1 \operatorname{tr}(G_T^{1/2}) + \frac{1}{2\eta} \left( D^2 \sqrt{\rho_{1:T}} + D^2 \operatorname{tr}(G_T^{1/2}) \right)$$

$$\leq D\sqrt{2\rho_{1:T}} + \left( \frac{C_1 + 1}{\sqrt{2}} \right) D \operatorname{tr}(G_T^{1/2})$$

$$= O(\operatorname{tr}(G_T^{1/2}) + \sqrt{\rho_{1:T}}).$$

Notably, the regret bound of Theorem 4.1 can be reformulated, as also presented in Feinberg et al. (2023).

**Corollary A.4.** *We define* $\lambda_{\tau:d}(G_T) = \sum_{i=\tau}^{d} \lambda_i(G_T)$*, where* $\lambda_i$ *is the i-th eigenvalue of* $G_T$*, we can rewrite the regret bound of FTSL as*

$$R(T) \leq O\left( \operatorname{tr}(G_T^{1/2}) + \sqrt{\lambda_{\tau:d}(G_T)} \right).$$

### A.3. Proof of Corollary A.4

We first give a lemma to give the upper bound of the cumulative $\rho_{1:T}$.

**Lemma A.5.** *(Lemma 1 in Feinberg et al. (2023)) The cumulative escaped masses* $\rho_{1:T}$ *in FD can be upper bounded as*

$$\rho_{1:T} \leq \min_{k=0,\ldots,\tau-1} \frac{\sum_{i=k+1}^{d} \lambda_i(G_T)}{\tau - k} \leq \sum_{i=\tau}^{d} \lambda_i(G_T) \stackrel{def}{=} \lambda_{\tau:d}(G_T),$$

where the last inequality is to set $k = \tau - 1$.

Combining Theorem 4.1 with Lemma A.5, we can get Corollary A.4.

### A.4. Proof of Theorem 4.2

We first introduce some guarantees of the fast frequent directions technique.

In Algorithm 3, we do not perform SVD on the sketching matrix $B_t$ every round. Instead, we maintain two matrices $M_t$ and $V_t$, which approximate the sketching matrix $B_t$ in Algorithm 2, that is

$$V_t M_t^{1/2} = B_t \in \mathbb{R}^{d \times 2\tau}.$$

In each round $t$, after receiving a new gradient $\mathbf{g}_t$, we first check whether this vector lies within the subspace spanned by $V_{t-1}$. If the vector is not contained within the subspace, we normalize it and subsequently add it to $V_{t-1}$, thereby enlarging the span of the subspace and ensuring $V_{t-1}V_{t-1}^{\top}\mathbf{g}_t = \mathbf{g}_t$. In our algorithm design, we want the matrix $V_t$ only contains orthonormal vectors, therefore we add $\frac{\mathbf{g}_t - \mathbf{g}'}{\|\mathbf{g}_t - \mathbf{g}'\|_2}$ to the first all zero column. In round $t$, we have

$$V_{t-1}M_{t-1}V_{t-1}^{\top} + \mathbf{g}_t\mathbf{g}_t^{\top} = V_{t-1}M_{t-1}V_{t-1}^{\top} + V_{t-1}V_{t-1}^{\top}\mathbf{g}_t\mathbf{g}_t^{\top}V_{t-1}V_{t-1}^{\top}$$

$$= V_{t-1}\left( M_{t-1} + V_{t-1}^{\top}\mathbf{g}_t\mathbf{g}_t^{\top}V_{t-1} \right)V_{t-1}^{\top}$$

$$= V_t\left( M_{t-1} + V_t^{\top}\mathbf{g}_t\mathbf{g}_t^{\top}V_t \right)V_t^{\top}.$$

According to Lemma A.3, we have

$$V_t M_t V_t^\top = B_t B_t^\top \preceq G_t.$$

In the following, we will prove $G_t \preceq \tilde{G}_t$, which is important for our analysis.

Assume we delete the eigenvalues of $M_k$ in round $k$, we calculate $\tilde{G}_k$ before we delete the eigenvalues of $M_k$ in our method. In round $k$, before we update $V_k$, we have $V_k = V_{k-1}$ and we let $V_k$ to represent the matrix before we delete its columns.

As we perform SVD to $V_k M_k^{1/2} \in \mathbb{R}^{d \times 2\tau}$, we delete the eigenvalues of $M_k$, update $V_k = V_k U_k$ and set the last $\tau + 1$ columns to zero. In round $k+1$, we have $\tilde{G}_{k+1} = V_{k+1}(\mathrm{diag}(\max\{\lambda_{[1:2\tau]}^{(k)} - \lambda_\tau^{(k)}, 0\}) + V_{k+1}^\top \mathbf{g}_{k+1}(V_{k+1}^\top \mathbf{g}_{k+1})^\top)V_{k+1}^\top + \rho_{1:k+1} I_{d \times d}$, and we can derive

$$
\begin{aligned}
&V_{k+1}(\mathrm{diag}(\max\{\lambda_{[1:2\tau]}^{(k)} - \lambda_\tau^{(k)}, 0\}) + V_{k+1}^\top \mathbf{g}_{k+1}(V_{k+1}^\top \mathbf{g}_{k+1})^\top)V_{k+1}^\top + \rho_{k+1} I_{d \times d} \\
=&V_{k+1}\mathrm{diag}(\max\{\lambda_{[1:2\tau]}^{(k)} - \lambda_\tau^{(k)}, 0\})V_{k+1}^\top + \lambda_\tau^{(k)} I_{d \times d} + \mathbf{g}_{k+1}\mathbf{g}_{k+1}^\top \\
\succeq&V_k U_k \mathrm{diag}(\max\{\lambda_{[1:2\tau]}^{(k)} - \lambda_\tau^{(k)}, 0\})U_k^\top V_k^\top + \lambda_\tau^{(k)}(V_k U_k)(V_k U_k)^\top + \mathbf{g}_{k+1}\mathbf{g}_{k+1}^\top \\
=&V_k U_k \mathrm{diag}(\max\{\lambda_{[1:2\tau]}^{(k)} - \lambda_\tau^{(k)}, 0\} + \lambda_\tau^{(k)})U_k^\top V_k^\top + \mathbf{g}_{k+1}\mathbf{g}_{k+1}^\top \\
\succeq&V_k U_k \mathrm{diag}(\lambda_{[1:2\tau]}^{(k)})U_k^\top V_k^\top + \mathbf{g}_{k+1}\mathbf{g}_{k+1}^\top \\
=&V_k M_k V_k^\top + \mathbf{g}_{k+1}\mathbf{g}_{k+1}^\top,
\end{aligned}
$$

where the second inequality is due to only first $\tau - 1$ columns of $\mathrm{diag}(\max\{\lambda_{[1:2\tau]}^{(k)} - \lambda_\tau^{(k)}, 0\})$ are non-zero, the first $\tau - 1$ columns of $V_{k+1}$ and $V_k U_k$ are same, and $U_k$ and $V_k$ are orthogonal matrices.

If we do not delete the eigenvalues of $M_k$ in round $k$, we have $\rho_{k+1} = 0$ and

$$
\begin{aligned}
V_{k+1} M_{k+1} V_{k+1} + \rho_{k+1} I_{d \times d} &= V_{k+1}(M_k + V_{k+1}^\top \mathbf{g}_{k+1}(V_{k+1}^\top \mathbf{g}_{k+1})^\top)V_{k+1}^\top \\
&= V_{k+1} M_k V_{k+1}^\top + \mathbf{g}_{k+1}\mathbf{g}_{k+1}^\top.
\end{aligned}
$$

Assume there are most $\ell$ eigenvalues in $M_k$, $\ell \leq 2\tau - 1$, and most $\ell + 1$ non-zero columns in $V_{k+1}$, most $\ell$ non-zero columns in $V_k$ and the first $\ell$ columns of $V_{k+1}$ and $V_k$ are same. We have the following

$$
\begin{aligned}
V_{k+1} M_{k+1} V_{k+1} + \rho_{k+1} I_{d \times d} &= V_{k+1} U_k \mathrm{diag}(\lambda_{[1:2\tau]}^{(k)})U_k^\top V_{k+1}^\top + \mathbf{g}_{k+1}\mathbf{g}_{k+1}^\top + 0 I_{d \times d} \\
&= V_k U_k \mathrm{diag}(\lambda_{[1:2\tau]}^{(k)})U_k^\top V_k^\top + \mathbf{g}_{k+1}\mathbf{g}_{k+1}^\top \\
&= V_k M_k V_k^\top + \mathbf{g}_{k+1}\mathbf{g}_{k+1}^\top,
\end{aligned}
\tag{7}
$$

where the second equality is due to only first $\ell$ elements in $\lambda_{[1:2\tau]}^{(k)}$ are non-zero and first $\ell$ columns of $V_k$ and $V_{k+1}$ are same.

Therefore, we have

$$\tilde{G}_t = V_t M_t V_t + \rho_{1:t} I_{d \times d} \succeq V_{t-1} M_{t-1} V_{t-1} + \rho_{1:t-1} I_{d \times d} + \mathbf{g}_t \mathbf{g}_t^\top = \tilde{G}_{t-1} + \mathbf{g}_t \mathbf{g}_t^\top.$$

By summing up, we can get

$$\tilde{G}_t \succeq \sum_{i=1}^t \mathbf{g}_i \mathbf{g}_i^\top = G_t.$$

Next, we need to ensure that after FFD, the preconditioning matrix $\tilde{G}_t$ is monotone. It is natural to verify that if we do not remove the eigenvalues of $M_t$, $\tilde{G}_t$ remains monotone.

Then, we will prove that $\tilde{G}_t$ remains monotone even if we delete the eigenvalues of $M_t$. Assume in round $k$, we delete the eigenvalues of $M_k$. In round $k$, the matrix $\tilde{G}_k = \rho_{1:k} I_{d \times d} + V_k U_k \mathrm{diag}(\lambda_{[1:2\tau]}^{(k)})U_k^\top V_k^\top$. And $\tilde{G}_{k+1} = \rho_{1:k+1} I_{d \times d} + V_{k+1} M_{k+1} V_{k+1}^\top$. In round $k + 1$, since we do not delete the eigenvalues of $M_{k+1}$, the first $\tau - 1$ columns of $V_{k+1}$

of round $k+1$ and $V_k U_k$ of round $k$ are same (Notably, $V_k$ is different in round $k$ and $k+1$). We have $M_{k+1} = \text{diag}(\max\{\lambda_{[1:2\tau]}^{(k)} - \lambda_\tau^{(k)}, 0\}) + (V_{k+1}^\top \mathbf{g}_{k+1})(V_{k+1}^\top \mathbf{g}_{k+1})^\top$. As we set $\rho_{k+1} = \lambda_\tau^{(k)}$, we can ensure

$$
\begin{aligned}
\tilde{G}_{k+1} &= \rho_{1:k+1} I_{d\times d} + V_{k+1} M_{k+1} V_{k+1}^\top \\
&= \rho_{1:k+1} I_{d\times d} + V_{k+1}(\text{diag}(\max\{\lambda_{[1:2\tau]}^{(k)} - \lambda_\tau^{(k)}, 0\}) + (V_{k+1}^\top \mathbf{g}_{k+1})(V_{k+1}^\top \mathbf{g}_{k+1})^\top) V_{k+1}^\top \\
&\succeq \rho_{1:k} I_{d\times d} + \lambda_\tau^{(k)} I_{d\times d} + V_{k+1} \text{diag}(\max\{\lambda_{[1:2\tau]}^{(k)} - \lambda_\tau^{(k)}, 0\}) V_{k+1}^\top \\
&\succeq \rho_{1:k} I_{d\times d} + V_k U_k \lambda_\tau^{(k)} I_{d\times d} U_k^\top V_k^\top + V_k U_k \text{diag}(\max\{\lambda_{[1:2\tau]}^{(k)} - \lambda_\tau^{(k)}, 0\}) U_k^\top V_k^\top \\
&\succeq \rho_{1:k} I_{d\times d} + V_k U_k \text{diag}(\lambda_{[1:2\tau]}^{(k)}) U_k^\top V_k^\top \\
&= \tilde{G}_k,
\end{aligned}
$$

where the second inequality is due to the $\tau$ to $2\tau$ columns of $\text{diag}(\max\{\lambda_{[1:2\tau]}^{(k)} - \lambda_\tau^{(k)}, 0\})$ is zero and the first $\tau - 1$ columns of $V_k U_k$ of round $k$ and $V_{k+1}$ of round $k+1$ are same, so we can replace $V_{k+1}$ with $V_k U_k$, and $V_k, U_k$ are orthogonal matrices.

Therefore, we can ensure $\tilde{G}_t$ is monotone in FTFSL.

Using the Eq (4), we have the following

$$
\sum_{t=1}^T f_t(\mathbf{x}_t) - \sum_{t=1}^T f_t(\mathbf{x}^*) \leq \underbrace{\frac{1}{\eta}\Psi_T(\mathbf{x}^*)}_{R_D} + \underbrace{\frac{\eta}{2}\sum_{t=1}^T \|\mathbf{g}_t\|_{\tilde{G}_{t-1}^{-1/2}}^2}_{R_G}.
$$

As for the term $R_D$, we can derive

$$
\begin{aligned}
R_D &= \frac{1}{\eta}\Psi_T(\mathbf{x}^*) = \frac{1}{2\eta}\left\langle \mathbf{x}^*, \tilde{G}_T^{1/2}\mathbf{x}^* \right\rangle \\
&= \frac{1}{2\eta}\left\langle \mathbf{x}^*, \left(V_T M_T V_T^\top + \rho_{1:T} I_{d\times d}\right)^{1/2}\mathbf{x}^* \right\rangle \\
&\leq \frac{1}{2\eta}\left\langle \mathbf{x}^*, \left(V_T M_T V_T^\top\right)^{1/2}\mathbf{x}^* \right\rangle + \frac{1}{2\eta}\left\langle \mathbf{x}^*, \left(\rho_{1:T} I_{d\times d}\right)^{1/2}\mathbf{x}^* \right\rangle \\
&\leq \frac{1}{2\eta}\left\|\left(\rho_{1:T} I_{d\times d}\right)^{1/2}\right\|\|\mathbf{x}^*\|^2 + \frac{1}{2\eta}\left\langle \mathbf{x}^*, \left(G_T\right)^{1/2}\mathbf{x}^* \right\rangle \\
&\leq \frac{1}{2\eta}D^2\sqrt{\rho_{1:T}} + \frac{1}{2\eta}D^2 \lambda_{\max}(G_T^{1/2}) \\
&\leq \frac{1}{2\eta}D^2\sqrt{\rho_{1:T}} + \frac{1}{2\eta}D^2 \text{tr}(G_T^{1/2}),
\end{aligned} \tag{8}
$$

where the second inequality is due to $V_T M_T V_T^\top = B_T B_T^\top \preceq G_T$ and we assume $\|\mathbf{x}^*\| \leq D$.

For the term $R_G$, we denote $b_t = \max\{\frac{\rho_{1:t-1}}{\|\mathbf{g}_t\|^2 + \rho_{1:t-1}}, 1\} \leq 1$, we have

$$
\begin{aligned}
\tilde{G}_{t-1} &= V_{t-1} M_{t-1} V_{t-1}^\top + \rho_{1:t-1} I_{d\times d} \\
&\succeq b_t(\|\mathbf{g}_t\|^2 I_{d\times d} + \rho_{1:t-1} I_d + B_{t-1} B_{t-1}^\top) \\
&\succeq b_t(\|\mathbf{g}_t\|^2 I_{d\times d} + G_{t-1}) \\
&\succeq b_t G_t.
\end{aligned}
$$

Then, letting $C_2 = \max_{t \in [T]} \frac{1}{\sqrt{b_t}}$, we can get

$$\sum_{t=1}^{T} \|\mathbf{g}_t\|_{\tilde{G}_{t-1}^{-1/2}}^2 = \sum_{t=1}^{T} \left\langle \mathbf{g}_t, (\tilde{G}_{t-1}^{1/2})^{-1} \mathbf{g}_t \right\rangle \le \sum_{t=1}^{T} \frac{1}{\sqrt{b_t}} \left\langle \mathbf{g}_t, (G_t^{1/2})^{-1} \mathbf{g}_t \right\rangle$$

$$\le C_2 \sum_{t=1}^{T} \left\langle \mathbf{g}_t, (G_t^{1/2})^{-1} \mathbf{g}_t \right\rangle \le C_2 \operatorname{tr}(G_T^{1/2}),$$

where the last inequality is due to Lemma A.2.

Therefore, we have

$$R_G = \frac{\eta}{2} \sum_{t=1}^{T} \|\mathbf{g}_t\|_{\tilde{G}_{t-1}^{-1/2}}^2 \le \eta C_2 \operatorname{tr}(G_T^{1/2}). \tag{9}$$

By combining inequalities (8) and (9), and setting $\eta = \frac{D}{\sqrt{2}}$, we can derive Theorem 4.2.

$$\begin{aligned}
R(T) &= \sum_{t=1}^{T} f_t(\mathbf{x}_t) - \sum_{t=1}^{T} f_t(\mathbf{x}^*) \\
&\le R_D + R_G \\
&\le \frac{1}{2\eta} D^2 \sqrt{\rho_{1:T}} + \frac{1}{2\eta} D^2 \operatorname{tr}(G_T^{1/2}) + \eta C_2 \operatorname{tr}(G_T^{1/2}) \\
&\le O(\operatorname{tr}(G_T^{1/2}) + \sqrt{\rho_{1:T}}).
\end{aligned}$$

### A.5. Proof of Theorem 4.3

According to the proof of Theorem 4.2, we have the following properties in FFD:

$$V_t M_t^{1/2} = B_t,$$
$$V_t U_t \Sigma_t U_t^\top V_t^\top \preceq G_t \preceq \tilde{G}_t.$$

Similar to the proof of Theorem 3 in Feinberg et al. (2023), we have the following lemma:

**Lemma A.6.** *Let $\{\mathbf{x}_t\}$ be the decision of Fast S-ADA and $\mathbf{x}^* \in \arg\min_{\mathbf{x} \in \mathbb{R}^d} \sum_{t=1}^{T} f_t(\mathbf{x})$, the regret bound of Fast S-ADA is*

$$R(T) \le \frac{1}{2\eta} \sum_{t=1}^{T} \|\mathbf{x}_t - \mathbf{x}^*\|_{\tilde{G}_t^{1/2} - \tilde{G}_{t-1}^{1/2}}^2 + \frac{\eta}{2} \sum_{t=1}^{T} \|\mathbf{g}_t\|_{\tilde{G}_t^{-1/2}}^2.$$

According to Lemma A.6, we have

$$R(T) \le \frac{1}{2\eta} \sum_{t=1}^{T} \|\mathbf{x}_t - \mathbf{x}^*\|_{\tilde{G}_t^{1/2} - \tilde{G}_{t-1}^{1/2}}^2 + \frac{\eta}{2} \sum_{t=1}^{T} \|\mathbf{g}_t\|_{\tilde{G}_t^{-1/2}}^2.$$

We first bound the term $\frac{\eta}{2} \sum_{t=1}^{T} \|\mathbf{g}_t\|_{\tilde{G}_t^{-1/2}}^2$, which is easier to bound than that in FTFSL, as we can directly apply a lemma on it.

$$\begin{aligned}
\sum_{t=1}^{T} \|\mathbf{g}_t\|_{\tilde{G}_t^{-1/2}}^2 &= \sum_{t=1}^{T} \left\langle \mathbf{g}_t, (\tilde{G}_t^{1/2})^{-1} \mathbf{g}_t \right\rangle \\
&\le \sum_{t=1}^{T} \left\langle \mathbf{g}_t, (G_t^{1/2})^{-1} \mathbf{g}_t \right\rangle \\
&\le 2 \operatorname{tr}(G_T^{1/2}),
\end{aligned}$$

where the fist inequality is due to $\tilde{G}_t^{-1} \preceq G_t^{-1}$ and the last inequality is due to Lemma A.2.

Next, we bound the term $\frac{1}{2\eta} \sum_{t=1}^{T} \|\mathbf{x}_t - \mathbf{x}^*\|_{\tilde{G}_t^{1/2} - \tilde{G}_{t-1}^{1/2}}^2$, which introduces the dependence on dimensionality $d$. We have

$$\frac{1}{2\eta} \sum_{t=1}^{T} \|\mathbf{x}_t - \mathbf{x}^*\|_{\tilde{G}_t^{1/2} - \tilde{G}_{t-1}^{1/2}}^2 \leq \frac{D_1^2}{2\eta} \operatorname{tr}\left(\tilde{G}_T^{1/2}\right),$$

where this inequality is due to monotonicity of $\tilde{G}_t$ and $D_1 = \max_{t \in [T]} \|\mathbf{x}_t - \mathbf{x}^*\|$.

Then we need to bound the term $\operatorname{tr}\left(\tilde{G}_T^{1/2}\right)$.

Assume we delete the eigenvalues of $M_k$ at round $k$. In round $k$, before we update $V_k$, $V_k = V_{k-1}$ and we use $V_k$ to represent the matrix before we delete its columns.

As we perform SVD to $V_k M_k^{1/2} \in \mathbb{R}^{d \times 2\tau}$, we denote $V_k^{1:\tau} \in \mathbb{R}^{d \times \tau}$ to be the first $\tau$ columns of $V_k U_k$ before we set the last $\tau + 1$ columns to $\mathbf{0}_d$, $N_k \in \mathbb{R}^{d \times (d-\tau)}$ be the complementary subspace of $V_k^{1:\tau}$, and the first $\tau - 1$ columns of $V_{k+1}$ and $V_k^{1:\tau}$ are same, $\rho_{k+1} = \lambda_\tau^{(k)}$ and $[V_k^{1:\tau}; N_k][V_k^{1:\tau}; N_k]^\top = I_{d \times d}$.

In round $k+1$, we have $\tilde{G}_{k+1} = V_{k+1}(\operatorname{diag}(\max\{\lambda_{[1:2\tau]}^{(k)} - \lambda_\tau^{(k)}, 0\}) + V_{k+1}^\top \mathbf{g}_{k+1}(V_{k+1}^\top \mathbf{g}_{k+1})^\top)V_{k+1}^\top + \rho_{1:k+1} I_{d \times d}$, and we can derive

$$\begin{aligned}
&V_{k+1}(\operatorname{diag}(\max\{\lambda_{[1:2\tau]}^{(k)} - \lambda_\tau^{(k)}, 0\}) + V_{k+1}^\top \mathbf{g}_{k+1}(V_{k+1}^\top \mathbf{g}_{k+1})^\top)V_{k+1}^\top + \rho_{k+1} I_{d \times d} \\
=&V_{k+1}\operatorname{diag}(\max\{\lambda_{[1:2\tau]}^{(k)} - \lambda_\tau^{(k)}, 0\})V_{k+1}^\top + \lambda_\tau^{(k)} I_{d \times d} + \mathbf{g}_{k+1}\mathbf{g}_{k+1}^\top \\
=&V_k U_k \operatorname{diag}(\max\{\lambda_{[1:2\tau]}^{(k)} - \lambda_\tau^{(k)}, 0\})U_k^\top V_k^\top + \lambda_\tau^{(k)}(V_k^{1:\tau}(V_k^{1:\tau})^\top + N_k N_k^\top) + \mathbf{g}_{k+1}\mathbf{g}_{k+1}^\top \\
\preceq&V_k U_k \operatorname{diag}(\max\{\lambda_{[1:2\tau]}^{(k)} - \lambda_\tau^{(k)}, 0\})U_k^\top V_k^\top + \lambda_\tau^{(k)}(V_k U_k(V_k U_k)^\top + N_k N_k^\top) + \mathbf{g}_{k+1}\mathbf{g}_{k+1}^\top \\
\preceq&V_k U_k \operatorname{diag}(\lambda_{[1:2\tau]}^{(k)})U_k^\top V_k^\top + \lambda_\tau^{(k)} N_k N_k^\top + \mathbf{g}_{k+1}\mathbf{g}_{k+1}^\top \\
=&\lambda_\tau^{(k)} N_k N_k^\top + V_k M_k V_k^\top + \mathbf{g}_{k+1}\mathbf{g}_{k+1}^\top \\
=&\rho_{k+1} N_k N_k^\top + V_k M_k V_k^\top + \mathbf{g}_{k+1}\mathbf{g}_{k+1}^\top,
\end{aligned}$$

where the second equality is due to only first $\tau - 1$ columns of $\operatorname{diag}(\max\{\lambda_{[1:2\tau]}^{(k)} - \lambda_\tau^{(k)}, 0\})$ are non-zero and the first $\tau - 1$ columns of $V_{k+1}$ and $V_k U_k$ are same, the first inequality is due to $V_k^{1:\tau}$ only contains $\tau$ orthogonal vectors at most.

If we do not delete the eigenvalues of $M_k$ in round $k$, we have $\rho_{k+1} = 0$ can derive

$$\begin{aligned}
V_{k+1} M_{k+1} V_{k+1} + \rho_{k+1} I_{d \times d} &= V_{k+1}(M_k + V_{k+1}^\top \mathbf{g}_{k+1}(V_{k+1}^\top \mathbf{g}_{k+1})^\top)V_{k+1}^\top \\
&= V_{k+1} M_k V_{k+1}^\top + \mathbf{g}_{k+1}\mathbf{g}_{k+1}^\top \\
&= V_{k+1} M_k V_{k+1}^\top + \mathbf{g}_{k+1}\mathbf{g}_{k+1}^\top \\
&= 0 N_k N_k^\top + V_{k+1} M_k V_{k+1}^\top + \mathbf{g}_{k+1}\mathbf{g}_{k+1}^\top \\
&= \rho_{k+1} N_k N_k^\top + V_{k+1} U_k \operatorname{diag}(\lambda_{[1:2\tau]}^{(k)})U_k^\top V_{k+1}^\top + \mathbf{g}_{k+1}\mathbf{g}_{k+1}^\top.
\end{aligned}$$

Assume there are most $\ell$ eigenvalues in $M_k$, $\ell \leq 2\tau - 1$, therefore, there are most $\ell + 1$ non-zero columns in $V_{k+1}$, most $\ell$ non-zero columns in $V_k$ and the first $\ell$ columns of $V_{k+1}$ and $V_k$ are same. We have the following

$$\begin{aligned}
V_{k+1} M_{k+1} V_{k+1} + \rho_{k+1} I_{d \times d} &= 0 N_k N_k^\top + V_{k+1} U_k \operatorname{diag}(\lambda_{[1:2\tau]}^{(k)})U_k^\top V_{k+1}^\top + \mathbf{g}_{k+1}\mathbf{g}_{k+1}^\top \\
&= 0 N_k N_k^\top + V_k U_k \operatorname{diag}(\lambda_{[1:2\tau]}^{(k)})U_k^\top V_k^\top + \mathbf{g}_{k+1}\mathbf{g}_{k+1}^\top \\
&= \rho_{k+1} N_k N_k^\top + V_k U_k \operatorname{diag}(\lambda_{[1:2\tau]}^{(k)})U_k^\top V_k^\top + \mathbf{g}_{k+1}\mathbf{g}_{k+1}^\top \\
&= \rho_{k+1} N_k N_k^\top + V_k M_k V_k^\top + \mathbf{g}_{k+1}\mathbf{g}_{k+1}^\top,
\end{aligned}$$
(10)

where the second equality is due to there are most $\ell$ non-zero elements in $\lambda_{[1:2\tau]}^{(k)}$ and first $\ell$ columns of $V_k$ and $V_{k+1}$ are same, and the third equality is due to $\rho_{k+1} = 0$.

Therefore, we have

$$V_{k+1} M_{k+1} V_{k+1} + \rho_{k+1} I_{d \times d} \preceq \rho_{k+1} N_k N_k^\top + V_k M_k V_k^\top + \mathbf{g}_{k+1} \mathbf{g}_{k+1}^\top.$$

By reduction, we have

$$\begin{aligned}
\tilde{G}_T &= V_T M_T V_T + \rho_{1:T} I_{d \times d} \\
&\preceq \rho_{1:T-1} I_{d \times d} + V_{T-1} M_{T-1} V_{T-1}^\top + \rho_T N_T N_T^\top + \mathbf{g}_T \mathbf{g}_T^\top \\
&\cdots \\
&\preceq \sum_{t=1}^T \rho_t N_t N_t^\top + \sum_{t=1}^T \mathbf{g}_t \mathbf{g}_t^\top \\
&= \sum_{t=1}^T \rho_t N_t N_t^\top + G_T.
\end{aligned}$$

Then we can rewrite the bound of $\text{tr}(\tilde{G}_T^{1/2})$ as $\text{tr}(G_T^{1/2}) + \text{tr}((\sum_{t=1}^T \rho_t N_t N_t^\top)^{1/2})$.

We just need to give bound of $\text{tr}((\sum_{t=1}^T \rho_t N_t N_t^\top)^{1/2})$. According to Corollary 4 in Feinberg et al. (2023), the upper bound of this term is

$$\text{tr}((\sum_{t=1}^T \rho_t N_t N_t^\top)^{1/2}) \leq \sqrt{d(d-\tau)\rho_{1:T}}.$$

Then we can derive the bound of $\text{tr}(\tilde{G}_T^{1/2})$.

$$\begin{aligned}
\text{tr}(\tilde{G}_T^{1/2}) &\leq \text{tr}(G_T^{1/2}) + \text{tr}((\sum_{t=1}^T \rho_t N_t N_t^\top)^{1/2}) \\
&\leq \text{tr}(G_T^{1/2}) + \sqrt{d(d-\tau)\rho_{1:T}}.
\end{aligned}$$

By setting $\eta = \frac{D_1}{\sqrt{2}}$, we have

$$\begin{aligned}
R(T) &= \sum_{t=1}^T f_t(\mathbf{x}_t) - \sum_{t=1}^T f_t(\mathbf{x}^*) \\
&\leq \frac{1}{2\eta} \sum_{t=1}^T \|\mathbf{x}_t - \mathbf{x}^*\|_{\tilde{G}_t^{1/2} - \tilde{G}_{t-1}^{1/2}}^2 + \frac{\eta}{2} \sum_{t=1}^T \|\mathbf{g}_t\|_{\tilde{G}_t^{-1/2}}^2 \\
&\leq \frac{D_1^2}{2\eta} (\text{tr}(G_T^{1/2}) + \sqrt{d(d-\tau)\rho_{1:T}}) + \eta \, \text{tr}(G_T^{1/2}) \\
&\leq O\left( \text{tr}(G_T^{1/2}) + \sqrt{d(d-\tau)\rho_{1:T}} \right).
\end{aligned}$$

### A.6. Proof of Theorem 4.4

In the following, we give the proof of Theorem 4.4.

Due to the update in Algorithm 5 is performed on the matrix space, it poses challenges for the analysis. Therefore, we first introduce an equivalent update in vector form.

Recall the update in Algorithm 5, $X_t = -\eta \tilde{L}_t^{-1/4} \overline{G}_t^X \tilde{R}_t^{-1/4}$. We define $\tilde{H}_t = \tilde{L}_t^{1/4} \otimes \tilde{R}_t^{1/4} \in \mathbb{R}^{mn \times mn}, \overline{L}_t = \hat{L}_t \hat{L}_t^\top \in \mathbb{R}^{m \times m}, \overline{R}_t = \hat{R}_t \hat{R}_t^\top \in \mathbb{R}^{n \times n}, \mathbf{g}_t = \overline{\text{vec}}(G_t^X)$ and $\mathbf{x}_t = \overline{\text{vec}}(X_t)$, where $\overline{\text{vec}}$ denotes the row-major vectorization of a given matrix. Kronecker product $\otimes$ obeys the following properties as shown in Gupta et al. (2018).

**Lemma A.7.** *(Lemma 3,4 in* Gupta et al. (2018)*) For matrices $A, A', B, B'$ of appropriate dimensions and vectors $\mathbf{x}, \mathbf{y}$, $L \in \mathbb{R}^{m \times m}$, $R \in \mathbb{R}^{n \times n}$, $G \in \mathbb{R}^{m \times n}$, the following properties hold:*

1. $(A \otimes B)(A' \otimes B') = (AA') \otimes (BB')$.

2. $(A \otimes B)^\top = A^\top \otimes B^\top$.

3. $A, B \succeq 0$, $(A \otimes B)^{-1} = A^{-1} \otimes B^{-1}$.

4. $A \succeq A'$, $B \succeq B'$, then $A \otimes B \succeq A' \otimes B'$.

5. $\text{tr}(A \otimes B) = \text{tr}(A) \text{tr}(B)$.

6. $\overline{\text{vec}}(\mathbf{x}\mathbf{y}^\top) = \mathbf{x} \otimes \mathbf{y}$.

7. $(L \otimes R^\top)\overline{\text{vec}}(G) = \overline{\text{vec}}(LGR)$.

By defining $\overline{\mathbf{g}}_t = \sum_{i=1}^{t} \mathbf{g}_i$, we can rewrite the update in Algorithm 5 as

$$\mathbf{x}_{t+1} = -\eta \tilde{H}_t^{-1} \overline{\mathbf{g}}_t,$$

which is equal to

$$\mathbf{x}_{t+1} = \underset{\mathbf{x} \in \mathbb{R}^{mn}}{\arg\min} \, \eta \langle \overline{\mathbf{g}}_t, \mathbf{x} \rangle + \frac{1}{2} \|\mathbf{x}\|_{\tilde{H}_t}^2.$$

As $\tilde{L}_t$ and $\tilde{R}_t$ is monotonically increasing with $t$, it is not hard to find that $\tilde{H}_t$ is also monotonically increasing with $t$. Thus, by using Lemma A.1, we have the similar inequality:

$$R(T) \leq \underbrace{\frac{1}{\eta} \Psi_T(\mathbf{x}^*)}_{R_D} + \underbrace{\frac{\eta}{2} \sum_{t=1}^{T} \|\mathbf{g}_t\|_{\tilde{H}_{t-1}^{-1}}^2}_{R_G}, \tag{11}$$

where $\Psi_T(\mathbf{x}^*) = \frac{1}{2} \langle \mathbf{x}^*, \tilde{H}_T \mathbf{x}^* \rangle$.

We first give the bound of $R_D$.

$$\Psi_T(\mathbf{x}^*) = \frac{1}{2} \langle \mathbf{x}^*, \tilde{H}_T \mathbf{x}^* \rangle \leq \frac{1}{2} \left\| \tilde{H}_T \right\| \|\mathbf{x}^*\|^2.$$

Then we introduce a lemma to give an equality about the norm of Kronecker product.

**Lemma A.8.** *(Theorem 8 in* Lancaster & Farahat (1972)*) For two matrices $A$ and $B$, the following equality holds*

$$\|A \otimes B\| = \|A\| \|B\|.$$

We have $\|\mathbf{x}^*\| = \|X^*\|_F \leq D_{\mathcal{M}}$. According to Lemma A.8, $\left\| \tilde{H}_T \right\| = \left\| \tilde{L}_T^{1/4} \otimes \tilde{R}_T^{1/4} \right\| = \left\| \tilde{L}_T^{1/4} \right\| \left\| \tilde{R}_T^{1/4} \right\|$, then we need to give the bound of $\left\| \tilde{L}_T^{1/4} \right\|$ and $\left\| \tilde{R}_T^{1/4} \right\|$, respectively. We first define $L_T = \sum_{t=1}^{T} (G_t^X)^\top G_t^X + \epsilon I_{m \times m}$ and $R_T = \sum_{t=1}^{T} (G_t^X)^\top G_t^X + \epsilon I_{n \times n}$. Using Lemma A.3, it is not hard to verify that $\overline{L}_t + \epsilon I_{m \times m} \preceq L_t$, $\overline{R}_t + \epsilon I_{n \times n} \preceq R_t$. Therefore, we have

$$\begin{aligned}
\left\| \tilde{L}_T^{1/4} \right\| &= \left\| (\overline{L}_T + \epsilon I_{m \times m} + \rho_{1:T}^L I_{m \times m})^{1/4} \right\| \\
&\leq \left\| (\overline{L}_T + \epsilon I_{m \times m})^{1/4} + (\rho_{1:T}^L I_{m \times m})^{1/4} \right\| \\
&\leq \left\| (\overline{L}_T + \epsilon I_{m \times m})^{1/4} \right\| + \left\| (\rho_{1:T}^L I_{m \times m})^{1/4} \right\| \\
&\leq \left\| (\overline{L}_T + \epsilon I_{m \times m})^{1/4} \right\| + (\rho_{1:T}^L)^{1/4} \\
&\leq \text{tr}((\overline{L}_T + \epsilon I_{m \times m})^{1/4}) + (\rho_{1:T}^L)^{1/4} \\
&\leq \text{tr}(L_T^{1/4}) + (\rho_{1:T}^L)^{1/4},
\end{aligned}$$

where the fourth inequality is due to have for positive semidefinite matrices $\operatorname{tr}(\cdot) \geq \|\cdot\|$ and last inequality is due to the monotonicity of $\operatorname{tr}(\cdot)$.

We also have

$$
\begin{aligned}
\left\|\tilde{R}_T^{1/4}\right\| &= \left\|(\overline{R}_T + \epsilon I_{n\times n} + \rho_{1:T}^R I_{n\times n})^{1/4}\right\| \\
&\leq \left\|(\overline{R}_T + \epsilon I_{n\times n})^{1/4} + (\rho_{1:T}^R I_{n\times n})^{1/4}\right\| \\
&\leq \left\|(\overline{R}_T + \epsilon I_{n\times n})^{1/4}\right\| + \left\|(\rho_{1:T}^R)^{1/4} I_{n\times n}\right\| \\
&\leq \left\|(\overline{R}_T + \epsilon I_{n\times n})^{1/4}\right\| + (\rho_{1:T}^R)^{1/4} \\
&\leq \operatorname{tr}((\overline{R}_T + \epsilon I_{n\times n})^{1/4}) + (\rho_{1:T}^R)^{1/4} \\
&\leq \operatorname{tr}(R_T^{1/4}) + (\rho_{1:T}^R)^{1/4}.
\end{aligned}
$$

Therefore, we can get

$$
R_D = \frac{1}{\eta} \Psi_T(\mathbf{x}^*) \leq \frac{D_{\mathcal{M}}^2}{2\eta} (\operatorname{tr}(L_T^{1/4}) + (\rho_{1:T}^L)^{1/4})(\operatorname{tr}(R_T^{1/4}) + (\rho_{1:T}^R)^{1/4}). \tag{12}
$$

To give the bound of $R_G$, we first introduce a lemma.

**Lemma A.9.** *(Lemma 8 in* Gupta et al. (2018)*) If $G_t^X \in \mathbb{R}^{m \times n}$ with rank at most $r$, and $\mathbf{g}_t = \overline{\operatorname{vec}}(G_t^X)$, then $\forall \epsilon \geq 0, \forall t$,*

$$
\epsilon I_{mn \times mn} + \frac{1}{r} \sum_{i=1}^t \mathbf{g}_i \mathbf{g}_i^\top \preceq \left(\epsilon I_{m\times m} + \sum_{i=1}^t G_i^X (G_i^X)^\top\right)^{1/2} \otimes \left(\epsilon I_{n\times n} + \sum_{i=1}^t (G_i^X)^\top (G_i^X)\right)^{1/2}.
$$

Then we utilize a lemma in Feinberg et al. (2023).

**Lemma A.10.** *(Lemma 14 in* Feinberg et al. (2023)*) Let $V_t \Sigma_t^L V_t^\top = \overline{L}_{t-1} + G_t^X (G_t^X)^\top$ be the eigendecomposition of the un-deflated sketch. We assume $\operatorname{rank}(\Sigma_t^L) = k, k \in [\tau - 1, \tau - 1 + r]$. Write $V_t = [V_t^*, V_t^\perp]$, where $V_t^*$ contains the first $k$ columns of $V_t$. And for the right conditioner $W_t \Sigma_t^R W_t^\top = \overline{R}_{t-1} + (G_t^X)^\top G_t^X$. Write $W_t = [W_t^*, W_t^\perp]$, where $W_t^*$ contains the first $k$ columns of $W_t$. Define $N_t^L = V_t^\perp (V_t^\perp)^\top$ and $N_t^R = W_t^\perp (W_t^\perp)^\top$, then we have*

$$
\widetilde{L}_t \succeq \sum_{i=1}^t G_i^X (G_i^X)^\top + \sum_{i=1}^t \rho_i^L N_i^L + \epsilon I_{m\times m} = M_t^L,
$$

$$
\widetilde{R}_t \succeq \sum_{i=1}^t (G_i^X)^\top G_i^X + \sum_{i=1}^t \rho_i^R N_i^R + \epsilon I_{n\times n} = M_t^R.
$$

According to Lemma A.3 and Lemma A.10, we have $M_t^L \succeq \epsilon I_{m\times m} + \sum_{i=1}^t G_i^X (G_i^X)^\top$ and $M_t^R \succeq \epsilon I_{n\times n} + \sum_{i=1}^t (G_i^X)^\top G_i^X$. Using Lemma A.7, we can derive

$$
I_{m\times m} \otimes \left(\epsilon I_{n\times n} + \sum_{i=1}^t (G_i^X)^\top G_i^X\right) \preceq I_{m\times m} \otimes M_t^R, \quad \left(\epsilon I_{m\times m} + \sum_{i=1}^t G_i^X (G_i^X)^\top\right) \otimes I_{n\times n} \preceq M_t^L \otimes I_{n\times n}.
$$

Therefore, we have

$$
(\epsilon I_{mn \times mn} + \frac{1}{r} \sum_{i=1}^t \mathbf{g}_i \mathbf{g}_i^\top)^{1/2} \preceq \left(M_t^L\right)^{1/4} \otimes \left(M_t^R\right)^{1/4} \preceq \tilde{L}_t^{1/4} \otimes \tilde{R}_t^{1/4} = \tilde{H}_t.
$$

Then we define $\widehat{H}_t = \left(r \epsilon I_{mn \times mn} + \sum_{i=1}^t \mathbf{g}_i \mathbf{g}_i^\top\right)^{1/2}$, and can have

$$
\widehat{H}_t \preceq \sqrt{r} \tilde{H}_t.
$$

We want to give the lower bound of $\hat{H}_{t-1}$. By defining $c_t = \frac{r\epsilon}{\|\mathbf{g}_t\|^2 + r\epsilon}$, we have

$$\hat{H}_{t-1}^2 = r\epsilon I_{mn \times mn} + \sum_{i=1}^{t-1} \mathbf{g}_i \mathbf{g}_i^\top$$

$$\succeq c_t(\|\mathbf{g}\|_t^2 I_{mn \times mn} + \sum_{i=1}^{t-1} \mathbf{g}_i \mathbf{g}_i^\top + r\epsilon I_{mn \times mn})$$

$$\succeq c_t(r\epsilon I_{mn \times mn} + \sum_{i=1}^{t} \mathbf{g}_i \mathbf{g}_i^\top)$$

$$= c_t \hat{H}_t^2.$$

Define $A_1 = \min_{t \in [T]}(\sqrt{c_t})$. We have $A_1 \hat{H}_t \preceq \hat{H}_{t-1} \preceq \sqrt{r}\tilde{H}_{t-1}$, which means $\frac{1}{\sqrt{r}}\tilde{H}_{t-1}^{-1} \preceq \hat{H}_{t-1}^{-1} \preceq \frac{1}{A_1}\hat{H}_t^{-1}$.

We can derive

$$\sum_{t=1}^{T} \|\mathbf{g}_t\|_{\tilde{H}_{t-1}^{-1}}^2 = \sum_{t=1}^{T} \left\langle \mathbf{g}_t, \tilde{H}_{t-1}^{-1}\mathbf{g}_t \right\rangle \leq \sqrt{r}\sum_{t=1}^{T} \left\langle \mathbf{g}_t, \hat{H}_{t-1}^{-1}\mathbf{g}_t \right\rangle$$

$$\leq \frac{\sqrt{r}}{A_1}\sum_{t=1}^{T} \left\langle \mathbf{g}_t, \hat{H}_t^{-1}\mathbf{g}_t \right\rangle.$$

Based on the proof of Theorem 5 in Feinberg et al. (2023), we obtain the following inequality.

$$\sum_{t=1}^{T} \left\langle \mathbf{g}_t, \hat{H}_t^{-1}\mathbf{g}_t \right\rangle \leq 2\operatorname{tr}(\hat{H}_T).$$

Therefore, we have

$$\sum_{t=1}^{T} \|\mathbf{g}_t\|_{\tilde{H}_{t-1}^{-1}}^2 \leq \frac{\sqrt{r}}{A_1}\sum_{t=1}^{T} \left\langle \mathbf{g}_t, \hat{H}_t^{-1}\mathbf{g}_t \right\rangle \leq \frac{2}{A_1}\sqrt{r}\operatorname{tr}(\hat{H}_T).$$

Then, we need to bound the term $\hat{H}_T$.

$$\hat{H}_t^2 = (r\epsilon I_{mn \times mn} + \sum_{i=1}^{T} \mathbf{g}_i \mathbf{g}_i^\top) \preceq r\left(\epsilon I_{m \times m} + \sum_{i=1}^{T} G_i^X (G_i^X)^\top\right)^{1/2} \otimes \left(\epsilon I_{n \times n} + \sum_{i=1}^{T} (G_i^X)^\top G_i^X\right)^{1/2} = rL_T^{1/2} \otimes R_T^{1/2},$$

which means $\hat{H}_t \preceq \sqrt{r}L_T^{1/4} \otimes R_T^{1/4}$.

By defining $C_3 = \frac{1}{A_1}$, we can derive the bound of $R_G$.

$$R_G = \frac{\eta}{2}\sum_{t=1}^{T} \|\mathbf{g}_t\|_{\tilde{H}_{t-1}^{-1}}^2 \leq \eta C_3 \sqrt{r}\operatorname{tr}(\hat{H}_T)$$

$$\leq \eta C_3 r \operatorname{tr}(L_T^{1/4})\operatorname{tr}(R_T^{1/4}).$$

By setting $\eta = \frac{D_\mathcal{M}}{\sqrt{r}}$, the final regret bound is

$$\sum_{t=1}^{T} f_t(\mathbf{x}_t) - f_t(\mathbf{x}^*) \leq \eta C_3 r \operatorname{tr}(L_T^{1/4})\operatorname{tr} R_T^{1/4} + \frac{D_\mathcal{M}^2}{2\eta}(\operatorname{tr}(L_T^{1/4}) + (\rho_{1:T}^L)^{1/4})(\operatorname{tr}(R_T^{1/4}) + (\rho_{1:T}^R)^{1/4})$$

$$\leq \sqrt{r}D_\mathcal{M}C_3 \operatorname{tr}(L_T^{1/4})\operatorname{tr}(R_T^{1/4}) + \frac{\sqrt{r}D_\mathcal{M}}{2}(\operatorname{tr}(L_T^{1/4}) + (\rho_{1:T}^L)^{1/4})(\operatorname{tr}(R_T^{1/4}) + (\rho_{1:T}^R)^{1/4}) \tag{13}$$

$$= O\left(\sqrt{r}(\operatorname{tr}(L_T^{1/4}) + (\rho_{1:T}^L)^{1/4})(\operatorname{tr}(R_T^{1/4}) + (\rho_{1:T}^R)^{1/4})\right).$$

# B. Online to Batch Reduction

---

**Algorithm 6** Online to Batch Conversion

---

1: **Input:** Time horizon $T$, rounds $N$, smoothness parameter $L$, OCO method $\mathcal{A}$
2: Initialize $\mathbf{x}_1$ to be any point in the domain
3: **for** $t = 1$ to $T$ **do**
4:     Construct $f_t(\mathbf{x}) = f(\mathbf{x}) + L \|\mathbf{x} - \mathbf{x}_t\|^2$
5:     Set $\mathbf{x}_t^1 = \mathbf{x}_t$ and pass $\mathbf{x}_t^1$ to $\mathcal{A}$
6:     **for** $i = 1$ to $N$ **do**
7:         Play $\mathbf{x}_t^i$, derive the gradient $\nabla f_t(\mathbf{x}_t; \xi_t)$, and construct $g_t^i(\mathbf{x}) = \nabla f_t(\mathbf{x}_t; \xi_t)^\top \mathbf{x}$
8:         Send $g_t^i(\mathbf{x})$ to $\mathcal{A}$ and receive $\mathbf{x}_t^{i+1}$
9:     **end for**
10:    Update $\mathbf{x}_{t+1} = \frac{1}{N} \sum_{i=1}^N \mathbf{x}_t^i$
11: **end for**
12: **Return** $\mathbf{x}_k = \arg\min_{k \in [T+1]} \|\nabla f(\mathbf{x}_t)\|$

---

In this section, we give some details for the reduction of non-convex stochastic optimization to online convex optimization for completeness. We use the framework of Agarwal et al. (2019), which Feinberg et al. (2023) also adopt before. Under this framework, we optimize a non-convex loss function $f(\mathbf{x})$ through constructing a new loss function $f_t(\mathbf{x}) = f(\mathbf{x}) + L \|\mathbf{x} - \mathbf{x}_t\|^2$, which is strongly convex. In each round, we pass the loss function $f_t(\mathbf{x})$ to any OCO method, $\mathcal{A}$, (it can be any algorithm in this paper), and use $\mathcal{A}$ to optimize it for $N$ rounds. When deriving the stochastic gradient, we use a batch $\xi_t$ to derive $\nabla f_t(\mathbf{x}_t; \xi_t)$, which satisfy $\mathbb{E}[\nabla f_t(\mathbf{x}_t; \xi_t)] = \nabla f_t(\mathbf{x}_t)$ and $\mathbb{E}[\|\nabla f_t(\mathbf{x}_t; \xi_t) - \nabla f_t(\mathbf{x}_t)\|^2] \leq \sigma^2$. The algorithm is stated in Algorithm 6, and we provide the convergence guarantees in the following. We first define the adaptive ratio.

**Definition B.1.** We denote $\mathbf{x}_\mathcal{A}$ be the output of an OCO method $\mathcal{A}$ and $\mathbf{x}^* \in \arg\min_{\mathbf{x} \in \mathbb{R}^d} f(x)$, we define the adaptive ratio of $\mathcal{A}$ as

$$\mu_A(f) = \frac{f(\mathbf{x}_A) - f(\mathbf{x}^*)}{\|\mathbf{x}_1 - \mathbf{x}^*\| \frac{\sigma}{T}}$$

Then we provide the convergence of this reduction.

**Theorem B.2.** *(Theorem A.2 in Agarwal et al. (2019)) We assume $f(\mathbf{x})$ is $L$-smooth, $\|\nabla^2 f(\mathbf{x})\| \leq L$, $\max_{\mathbf{x},\mathbf{y}} f(\mathbf{x}) - f(\mathbf{y}) \leq F$, $\mathbb{E}[\|\nabla f_t(\mathbf{x}_t; \xi_t) - \nabla f_t(\mathbf{x}_t)\|^2] \leq \sigma^2$, and $\mu = \max_t \mu_\mathcal{A}(f_t)$. By setting $T = \frac{12ML}{\epsilon^2}$ and $N = \frac{48\mu^2\sigma^2}{\epsilon^2}$, the output of Algorithm 6 satisfies*

$$\mathbb{E}\left[\|\nabla f(\mathbf{x}_t^*)\|\right] \leq \epsilon.$$

It is evident that the total number of queries to the stochastic gradient oracle is $O(\mu^2\sigma^2/\epsilon^4)$.

By using this framework, we can translate the regret bound of an OCO algorithm into convergence guarantees for stochastic optimization.

# C. Full experiments

In this section, we conduct empirical studies to evaluate our proposed algorithms. In online classification task, we compare our methods with ADA-DIAG (Duchi et al., 2011), RADAGRAD (Krummenacher et al., 2016), FD-SON (Luo et al., 2018), ADA-FFD under two frameworks (Wan & Zhang, 2022) and S-ADA (Feinberg et al., 2023). In image classification task and language modeling task, we compare our methods with ADA-DIAG, ADA-FFD under two frameworks, S-ADA, Shampoo (Gupta et al., 2018), S-Shampoo (Feinberg et al., 2023). When it comes to hyper-parameter tuning, we either set the hyper-parameters as recommended in the original papers or tune them by grid search. For example, for learning rate $\eta$ and regularizer parameter $\epsilon$, we search them from the set $\{1e-5, 1e-4, 1e-3, 1e-2, 1e-1, 1, 5\}$ and $\{1e-5, 1e-4, 1e-3, 1e-2, 1e-1, 1, 5\}$, respectively, and select the best one. All experiments are conducted on 8 NVIDIA 3090 GPUs.

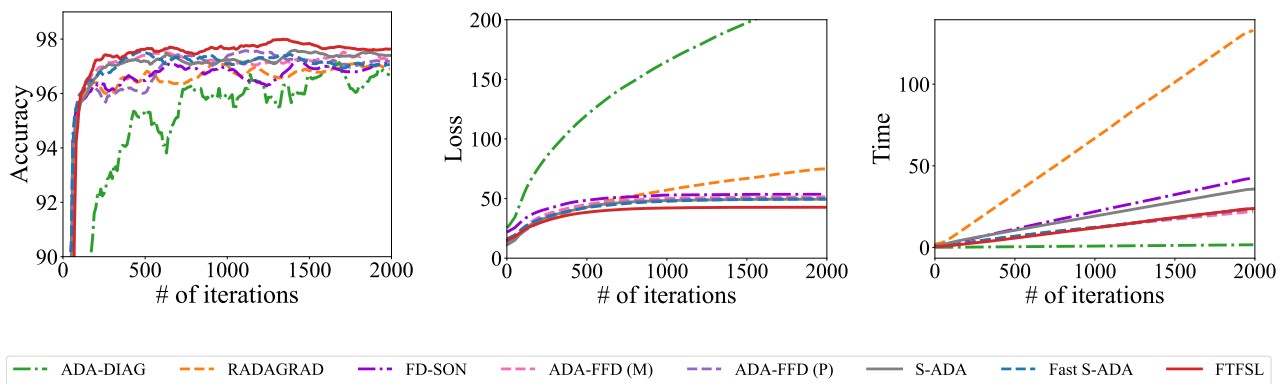

*Figure 3.* Results for Gisette dataset.

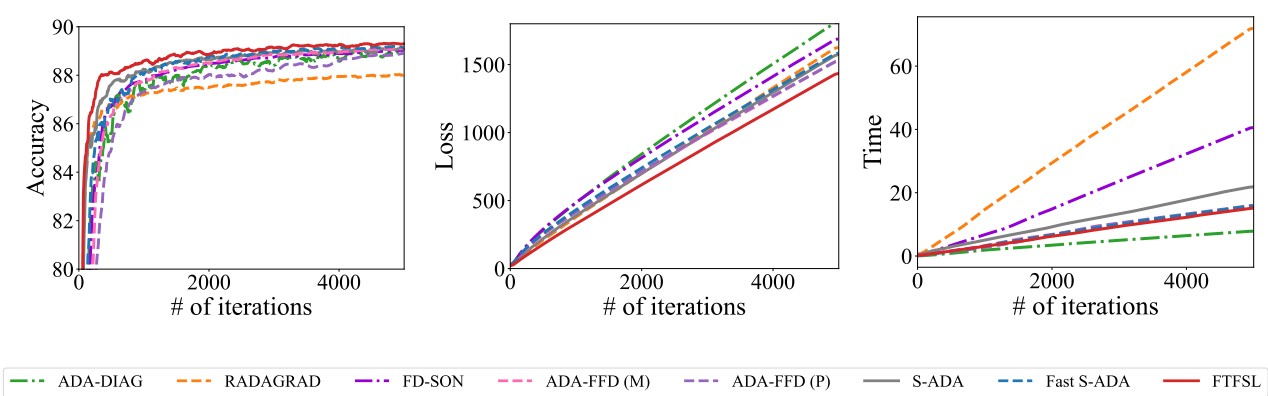

*Figure 4.* Results for Epsilon dataset.

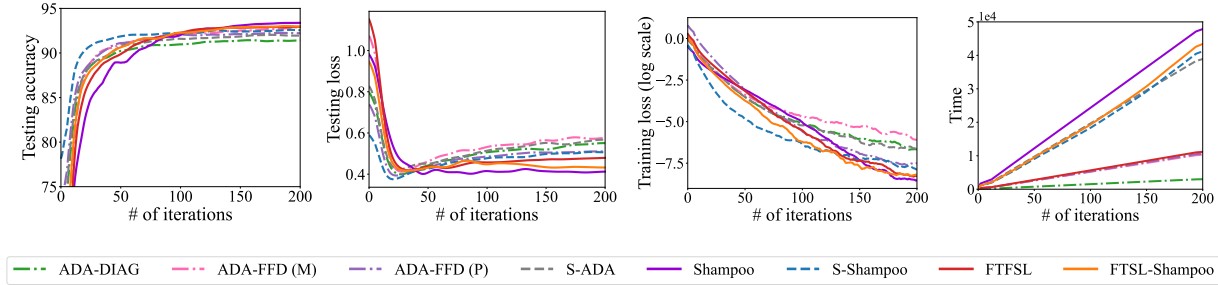

*Figure 5.* Results for CIFAR-10 dataset.

### C.1. Online Classification

First, we perform online classification to evaluate the performance of our methods with two real world datasets from LIBSVM (Chang & Lin, 2011) repository: Gisette and Epsilon, which are high-dimensional and dense. Particularly, Gisette contains 6000 training samples and 1000 testing samples, with 5000 features. Epsilon dataset consists of $400,000$ training samples and $100,000$ testing samples, with 2000 features. Let $T$ denote the number of total rounds. In each round $t \in [T]$, a batch of training examples $\{(\mathbf{w}_{t,1}, y_{t,1}), \ldots, (\mathbf{w}_{t,n}, y_{t,n})\}$ arrive, where $(\mathbf{w}_{t,i}, y_{t,i}) \in [-1,1]^d \times \{-1,1\}, i = 1, \ldots, n$. The online learner aims to predict a linear model $\mathbf{x}_t$ and suffers the hinge loss $f_t(\mathbf{x}_t) = \frac{1}{n} \sum_{i=1}^{n} \max\{0, 1 - y_t \mathbf{x}_t^\top \mathbf{w}_{t,i}\}$.

**Setup.** For Gisette, we set the batch size $n = 32$, the $\tau = 50$ to be $1\%$ of the original dimensionality, and $T = 2000$. For

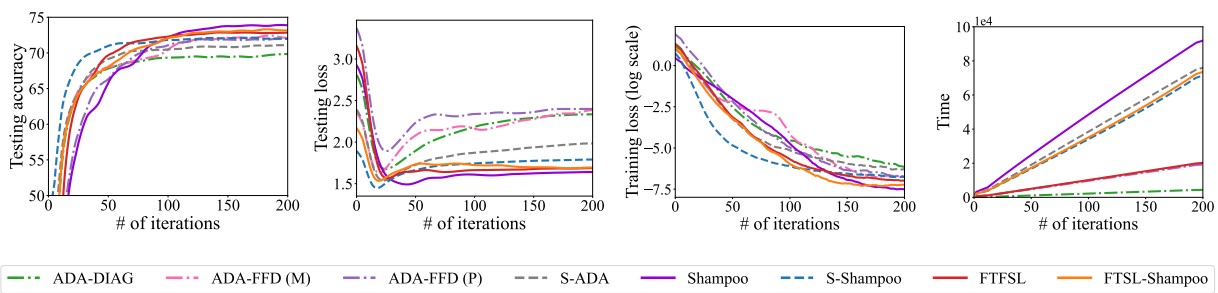

*Figure 6.* Results for CIFAR-100 dataset.

Epsilon, we set the batch size $n = 128$, the $\tau = 20$ and $T = 5000$ to pass through all the training data.

**Results.** Following Duchi et al. (2011), we adopt the performance of accuracy on the testing data to compare different methods. To better demonstrate the improvements of our methods, we additionally plot the loss and runtime (measured in seconds) of various methods. From Figure 3 and Figure 4, we observe that FTFSL outperforms all other methods in both loss and testing accuracy, aligning with its superior regret bound. Moreover, FTFSL and Fast S-ADA exhibit significantly lower runtimes compared to S-ADA, owing to their superior time complexities.

### C.2. Image Classification

In this section, we conduct numerical experiments on multi-class image classification tasks to evaluate the performance of the proposed methods, we compare FTFSL and FTSL-Shampoo with several baseline methods. The experiments involve training ResNet18 and ResNet34 models (He et al., 2016) on the CIFAR-10 and CIFAR-100 datasets (Krizhevsky, 2009), respectively, for 200 iterations with batch size of 128.

**Setup.** For ADA-FFD, S-ADA, FTFSL, the sketching size $\tau$ is determined based on the dimensionality of the flattened gradient, which is defined as:

$$\tau = \min\{\lceil d \times 0.1 \rceil, 100\},$$

where $d$ represents the total elements of parameters in each layer. We dynamically set the upper bound of the sketching size based on the dimensionality of each layer. For S-Shampoo and FTSL-Shampoo, due to its memory efficiency, we set $\tau = \lceil 0.1 \times d_i \rceil$, where $d_i$ is the dimensionality of the $i$-th dimension of a gradient. For the sake of fairness, we do not employ momentum trick.

**Results.** We plot the loss value and the accuracy against the iterations on the CIFAR-10 and CIFAR-100 in Figure 5 and Figure 6, respectively. It is observed that, for training loss and testing accuracy, our FTSL-Shampoo achieves comparable performance with respect to Shampoo, while significantly improving memory efficiency and reducing running time, which aligns with the theoretical guarantees. Additionally, our FTFSL converges more quickly than other sketching based algorithms, indicating the effectiveness of the proposed method. Moreover, we also present the running time of each method. FTFSL demonstrates a significant reduction in running time compared to S-ADA, owing to its improved time complexity.

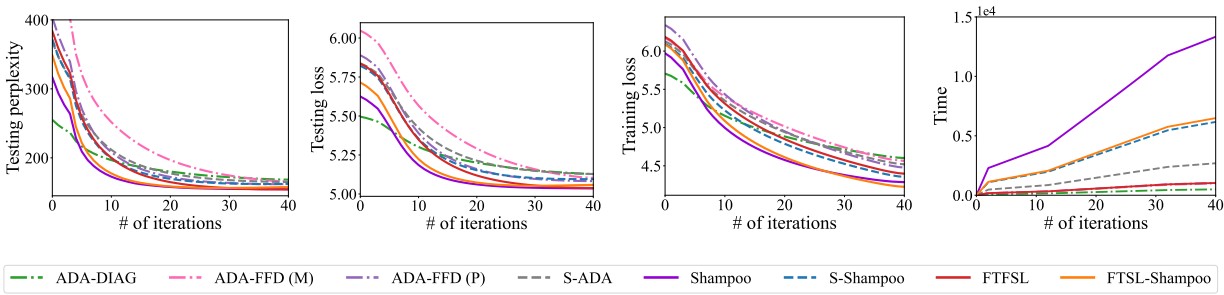

*Figure 7.* Results for WikiText-2 dataset.

### C.3. Language Modeling Task

In this section, we perform experiments on language modeling task. Concretely, we train a 2-layer Transformer (Vaswani et al., 2017) over the WiKi-Text2 dataset (Merity, 2016). We use 256 dimensional word embeddings, 256 hidden unites and 2 heads. We also clip the gradients by norm $0.5$ in case of the exploding gradient. The batch size is set as 64 and all methods are trained for 40 epochs with dropout rate 0.1.

**Setup.** The experimental setup follows that of image classification. For computational efficiency, we do not employ a preconditioning matrix in the embedding layer.

**Results.** We report the loss, perplexity and the run time in Figure 7. As can be seen, FTFSL and FTSL-Shampoo suffer lower loss and obtain better perplexity compared to other sketching based algorithms, indicating the effectiveness of the proposed methods. Moreover, FTFSL exhibits markedly improved efficiency over S-ADA.

