# OpenReview forum: "Dimension-Free Adaptive Subgradient Methods with Frequent Directions"
_ICML.cc/2025/Conference — ICML 2025 poster_

### Official Review · Reviewer_KDEM · 2025-03-09

**Overall Recommendation:** 2

**Summary:**

In machine learning, the seminal work [DHS'11] proposed the adaptive subgradient method with full matrices (ADA-FULL), which requires maintaining a preconditioning matrix with $O(d^2)$ space and $O(d^3)$ running time. However, ADA-FULL suffers from high-dimensional dependence in its regret bound and computational complexity, making it inefficient for large-scale optimization problems. To address these limitations, several methods have been proposed, including ADA-FD [WZ'18] and S-ADA [FCSAH'23], which leverage matrix sketching techniques to reduce computational costs. This paper further advances the acceleration of adaptive subgradient methods by integrating Frequent Directions (FD), a powerful matrix sketching approach. Specifically, it introduces the Follow-the-Sketchy-Leader (FTSL) method, which improves regret bound, space efficiency, or time complexity compared to prior works. Additionally, the paper extends these techniques to Shampoo, a second-order optimization method, resulting in the FTSL-Shampoo algorithm, which enhances memory efficiency and achieves dimension-free theoretical guarantees. Experiments on real datasets for online classification and image classification tasks demonstrate that the proposed methods outperform existing approaches in terms of test accuracy and running time.

**Claims And Evidence:**

I believe so, though I haven’t carefully reviewed the proofs in the appendix.

**Essential References Not Discussed:**

None.

**Experimental Designs Or Analyses:**

Yes, I reviewed the experiments in Section 5 and Appendix D. In online classification, the proposed methods are compared with prior works on the datasets Gisette and Epsilon of LIBSVM; in image classification, the experiments are implemented on CIFAR-10 and CIFAR-100.

**Methods And Evaluation Criteria:**

Yes, this paper compares with prior works on two tasks: online classification and image classification, in terms of test accuracy, training or test loss, and running time.

**Other Comments Or Suggestions:**

None.

**Other Strengths And Weaknesses:**

$\textbf{Strengths:}$ (1) This paper combines FD with ADA-FULL and proposes some new methods, including FTSL, FTFSL, and FTSL-Shampoo. On the theoretical aspect, these new methods improve the existing works with respect to regret bound, space or running time.
(2) The experimental results indicate the efficiency of the proposed methods on two real tasks.

$\textbf{Weaknesses:}$ (1) In Table 1, the proposed method FTSL has the same space and time complexity with the method ADA-FD (P), and their regret bounds are quite close due to $\sqrt{\sum_{t=1}^{T} \rho_t} \le \sum_{t=1}^{T} \sqrt{\rho_t} \le \sqrt{T \sum_{t=1}^{T} \rho_t}$. Likewise, in Table 2, FTFSL and ADA-FFD (P) have the same situation. Consequently, from a theoretical perspective, the proposed methods FTSL and FTFSL do not offer a clear advantage over existing works.
(2) From a technical perspective, both this paper and the works ADA-FD/ADA-FFD adopt the framework that integrates FD with ADA-FULL, which somewhat limits the novelty of this paper.
(3) For the experiments, in Figures 3 and 4, the proposed method FTFSL doesn't have a significant advantage over other methods, like ADA-FFD (M); similarly, in Figures 5 and 6, the proposed FTSL-Shampoo method exhibits performance comparable to the prior methods Shampoo and S-Shampoo.
(4) This paper evaluates the proposed methods on only four real datasets for online classification and image classification, which is relatively limited and weakens the assessment of their effectiveness.

**Questions For Authors:**

See Weaknesses.

**Relation To Broader Scientific Literature:**

This paper integrates FD with Shampoo to develop FTSL-Shampoo, which provides improved efficiency for optimization problems with matrix variables (e.g., neural network training).

**Theoretical Claims:**

I didn't read the proofs in the appendix.

---

> ### Author Rebuttal · Authors · 2025-04-01
>
> Many thanks for your constructive feedback!
>
> ---
>
> Q1. FTSL and FTFSL have the same complexities with ADA-FD(P) and ADA-FFD(P), and regret bounds are close.
>
> A1. We acknowledge that the proposed methods have the same time and space complexities with ADA-FD(P) and ADA-FFD(P). However, we want to clarify that our regret bounds have a significant improvement compared to them.
> * The additional error terms in ADA-FD(P)/ADA-FFD(P) and FTSL/FTFSL are $\sum_{t=1}^T \sqrt{\rho_{t}}$ and $\sqrt{\sum_{t=1}^T\rho_{t}}$, respectively.
> * As you pointed out, we have $\sqrt{\sum_{t=1}^T\rho_{t}}\leq\sum_{t=1}^T\sqrt{\rho_{t}}\leq\sqrt{T\sum_{t=1}^T\rho_{t}}$, which means that there exists‌ a large gap of $\sqrt{T}$ in the worst case. Note that $T$ is the number of iterations, which keeps on increasing, and cannot be ignored.
> * Moreover, as discussed in Observation 2 of [FCSAH'23], the regret bound of ADA-FD(P)/ADA-FFD(P) is $\Omega(T^{3/4})$ in some cases, while FTSL/FTFSL achieves a better $O(T^{1/2})$ bound, providing a substantial advantage. We provide the proof below.
>
> According to [FCSAH'23], there exists a situation, we receive the linear loss $f_t(\textbf{x}) = \langle\textbf{x},\textbf{g}_t \rangle$, where $\textbf{g} _t \in \mathbb{R}^d$ is a random vector drawn iid from any distribution over $r\leq d$ orthonormal vectors.
>
> As pointed out by [FCSAH'23], for any $\tau\leq r$, the bound on the expected regret of ADA-FD(P)/ADA-FFD(P) is $\Omega(T^{3/4})$.
>
> The regret of FTSL/FTFSL is $\eta \text{tr}(G_T^{1/2})+\frac{1}{\eta}\sqrt{\sum _{t=1}^T\rho _{t}} \leq \eta \text{tr}(G_T^{1/2})+\frac{1}{\eta}\sqrt{T\max _{t\in [T]}\rho_t} \leq O(T^{1/2})$, where the last inequality is due to $\text{tr}(G_T^{1/2}) =O(T^{1/2}), \rho_t = 0$ or $1$, and setting $\eta = O(1)$.
>
> ---
>
> Q2. Both this paper and ADA-FD/ADA-FFD integrate FD with ADA-FULL, which limits the novelty.
>
> A2. While both our paper and ADA-FD/ADA-FFD integrate FD with ADA-FULL, there do exist significant differences from them.
>
> * Compared to ADA-FD and ADA-FFD, our algorithm tracks the information discarded in FD and incorporates it back into the preconditioning matrix, achieving a better regret bound.
> * Our work additionally considers a more practical setting, optimization problems with matrix variables. We integrate FD with Shampoo and provide a novel analysis under the primal-dual framework. The dimension-free theoretical guarantee is another contribution of this work.
>
> ---
>
> Q3. In Figures 3 and 4, FTFSL doesn't have a significant advantage over ADA-FFD (M). In Figures 5 and 6, FTSL-Shampoo exhibits performance comparable to Shampoo and S-Shampoo.
>
> A3. We believe there might be some misunderstandings in this part.
> * Actually, in Figures 3 and 4, FTFSL (red line) outperforms ADA-FFD (M) (pink dashed line) in terms of testing accuracy and training loss.
> * We want to clarify that FTSL-Shampoo is an approximate version of Shampoo, aiming to enhance computational efficiency while maintaining comparable effectiveness. FTSL-Shampoo utilizes less information in each round for updates (some eigenvalues are discarded). In Figures 5 and 6, FTSL-Shampoo exhibits performance comparable to Shampoo, while significantly improving memory efficiency and reducing running time, which aligns with the theoretical guarantees.
> * As demonstrated by the experimental results, FTSL-Shampoo (orange line) substantially outperforms S-Shampoo (blue dashed line) across all metrics, including testing accuracy, testing loss, and training loss.
>
> ---
>
> Q4. This paper evaluates methods on only four datasets.
>
> A4. Following your suggestion, we conduct experiments on an NLP task, and the results can be found at the following anonymous link: https://anonymous.4open.science/r/ICML-14491/results.pdf.
>
> Concretely, we train a 2-layer Transformer over the WiKi-Text2 dataset. We use 256 dimensional word embeddings, 256 hidden unites and 2 heads. The batch size is set as 64 and all methods are trained for 40 epochs with dropout rate 0.1.  As can be seen, FTFSL and FTSL-Shampoo suffer lower loss and obtain better perplexity compared to other sketching based algorithms, indicating the effectiveness of the proposed methods.
>
> Additionally, we would like to take this opportunity to clarify that the primary contribution of this paper lies in the theoretical aspects, which includes the following three key points:
> * First, we propose FTSL, which achieves a dimension-free regret bound and maintains the same memory complexity as previous works.
> * Second, we develop Fast S-ADA and FTFSL to further reduce the time complexity, while preserving the same regret bounds.
> * Next, we investigate optimization problems with matrix variables, a scenario commonly encountered in deep learning tasks. FTSL-Shampoo enjoys an enhanced theoretical guarantee than S-Shampoo [FCSAH'23].

---

### Official Review · Reviewer_GoSK · 2025-03-18

**Overall Recommendation:** 4

**Summary:**

This work proposed an adaptive online subgradient method with frequent directions. The main contribution is the regret bound is dimension-free and the algorithm only requires the time complexity of $O(\tau d)$ in each iteration.

## update after rebuttal
All of my questions have been addressed. Hence, I would like to increase my overall rating.

**Claims And Evidence:**

Yes.

**Essential References Not Discussed:**

See the part "Questions For Authors".

**Experimental Designs Or Analyses:**

The experimental designs and results sound reasonable.

**Methods And Evaluation Criteria:**

Yes.

**Other Comments Or Suggestions:**

See the part "Questions For Authors".

**Other Strengths And Weaknesses:**

See the part "Questions For Authors".

**Questions For Authors:**

The main comments:
1. The literature review should be more appropriate. The idea of adaptive FD by adding the cumulative discarded information of FD back was first proposed by Luo et al. (2019) and Chen et al. (2020), and their time complexity have achieved $O(\tau d)$ by using the similar trick in Section 4.2. The main contribution of Feinberg et al. (2023) is improving the regret bound of Luo et al. (2019).
2. The comparison with Spectral Compensation Frequent Directions (SCFD) (Chen et at., 2020) should be discussed. It seems that both this paper and Feinberg’s et al. (2023) method use SCFD to approximate $G_T$.
3. Assumption 3.3 looks very strong although it is introduced in previous work. Is it possible to replace the low-rank assumption with an approximately low-rank assumption?
4. Can we guarantee $\tilde G_T$ be non-singular during iterations?
5. The experiments test the algorithms on nonsmooth hinge loss, and I find the analysis also does not rely on the smoothness of the loss function. Therefore, the word “gradient” in many sentences should be replaced with “subgradient”. The notation $\nabla f_t$ is also somewhat inappropriate.

The minor comments:
1. The font of vec in Lemma B.7 should be consistent to the description before this lemma.
2. The domain of $\bf x$ is required in line 1051.

**Relation To Broader Scientific Literature:**

See the part "Questions For Authors".

**Theoretical Claims:**

The theoretical claims sound correct.

---

> ### Author Rebuttal · Authors · 2025-04-01
>
> Thank you for your valuable feedback!
>
> ---
>
> Q1. The literature review should be more appropriate.
>
> A1. Thank you for bringing these related works to our attention. After checking the papers, we acknowledge that the idea of adaptive FD by incorporating the cumulative discarded information is first introduced by Luo et al. (2019) and Chen et al. (2020). We sincerely apologize for the omission of these work and will include a discussion of them in the revised version.
>
> ---
>
> Q2. The comparison with SCFD (Chen et al., 2020).
>
> A2. Although our sketching technique is similar to SCFD, the setting and algorithmic design of our work are fundamentally *distinct*:
> * Chen et al. (2020) focus on linear contextual bandits, while our work investigates the *general online convex optimization* problem.
> * Their method is based on LinUCB, whereas our FTSL/FTFSL and FTSL-Shampoo are based on ADA-FULL and Shampoo, respectively.
> * Due to the *essential distinctions* in settings, our algorithmic design and theoretical analysis are different from Chen et al. (2020).
>
> ---
>
> Q3. Is it possible to replace the low-rank assumption with an approximately low-rank assumption?
>
> A3. First, we would like to emphasize that Assumption 3.3 is _only_ used in the analysis of FTSL-Shampoo. Moreover, the analyses of Shampoo and S-Shampoo also utilize this assumption. In our paper, Assumption 3.3 is used in Lemma B.9 to give the lower bounds of the sketching preconditioning matrices.
>
> Second, we can replace Assumption 3.3 with the approximately low-rank assumption. However, it would introduce an additional approximation term in the final regret, leading to a difference from the bounds of Shampoo and S-Shampoo.  We consider this modification as a direction for future research.
>
> ---
>
> Q4. Can we guarantee $\tilde{G}_t$ be non-singular during iterations?
>
> A4. In fact, $\tilde{G}_t$ is not always non-singular. It is _singular_ in the early stages of the iteration. According to Step 8 of Algorithm 2, it becomes _non-singular_ after a certain number of iterations. When $\tilde{G}_t$ is singular, we use the Moore-Penrose pseudoinverse in our analysis. In practice, we can ensure its non-singularity by adding a small regularization term $\epsilon I_d, \epsilon > 0$ into $\tilde{G}_t$.
>
> ---
>
> Q5. The word “gradient” in many sentences should be replaced with “subgradient”.
>
> Q6. Minor comments.
>
> A5 & A6. Thank you for pointing out these typos. We will correct this misuse and some minor typos in the revised version.
>
> **Reference:**
>
> Luo Luo, Cheng Chen, Zhihua Zhang, Wu-Jun Li, and Tong Zhang. Robust frequent
> directions with application in online learning. JMLR, 2019.
>
> Cheng Chen, Luo Luo, Weinan Zhang, Yong Yu, and Yijiang Lian. Efficient and robust high-dimensional linear contextual bandits. IJCAI, 2020.

---

> > ### Comment · Reviewer_GoSK · 2025-04-01
> >
> > Thanks for your detailed response. All of my questions have been addressed. Hence, I would like to increase my overall rating.

---

> > > ### Author Response · Authors · 2025-04-02
> > >
> > > Dear Reviewer GoSK,
> > >
> > > Thank you for your kind response! We will improve our paper according to your constructive reviews.
> > >
> > > Best regards,
> > >
> > > Authors

---

### Official Review · Reviewer_5FxL · 2025-03-31

**Overall Recommendation:** 3

**Summary:**

The paper proposes adaptive subgradient methods for online convex optimization that have better regret bounds and time complexities than existing methods. This is achieved by analyzing the frequent directions in the primal-dual framework.

**Claims And Evidence:**

The claims are supported by clear evidence.

**Essential References Not Discussed:**

Not to my knowledge.

**Experimental Designs Or Analyses:**

I have gone through the experiment designs and analyses and found them to be in order.

**Methods And Evaluation Criteria:**

The proposed evaluations seem standard and make sense. They showed some advantages of the proposed approach.

**Other Comments Or Suggestions:**

See the comments above.

**Other Strengths And Weaknesses:**

Strengths:
- Improved regret bounds and time complexities over existing methods.
- Numerical results show the efficacy of the proposed methods.

Weaknesses:
- As mentioned earlier, the non-smoothness of the loss functions should be handled more carefully in the analysis.
- The work makes heavy use of existing techniques. It will be good to explain how the new techniques developed in this paper have applications in other settings.

**Questions For Authors:**

See the "Other Strengths And Weaknesses" section.

**Relation To Broader Scientific Literature:**

The paper makes use of a number of ideas from the literature, including frequent directions, the primal-dual framework, and adaptive methods in online convex optimization. The results are obtained by combining these elements in a new manner.

**Theoretical Claims:**

I have gone through the proofs quickly, and they look mostly fine. Since the loss functions are only assumed to be convex and hence can be non-smooth, the paper should be careful not to mix gradients and subgradients and clearly indicate whether their arguments work for every subgradient or just one particular subgradient in the subdifferential.

---

> ### Author Rebuttal · Authors · 2025-04-01
>
> Thanks for your constructive comments!
>
> ---
>
> Q1: Since the loss functions are only assumed to be convex and hence can be non-smooth, the paper should be careful not to mix gradients and subgradients and clearly indicate whether their arguments work for every subgradient or just one particular subgradient in the subdifferential.
>
>
> A1: Thank you for pointing out this misuse. Actually, the analysis of FTSL does not require the smoothness. We utilize the convexity of the loss functions in Lemma B.1, which uses the subgradients and works for every subgradient. We will correct this misuse in the revised version.
>
> ---
>
> Q2: The work makes heavy use of existing techniques. It will be good to explain how the new techniques developed in this paper have applications in other settings.
>
>
> A2: Thank you for your suggestion. We present two potential applications of the new technique:
> * **LLM fine-tuning.** Our sketching technique can be incorporated into LLM fine-tuning. For example, parameter-efficient fine-tuning (PEFT) methods (Zhao et al., 2024) update models for each task within multiple subspaces. We can use this technique to merge them into a single subspace, making the updates more stable.
> * **Bandit problem.** Our sketching technique can also be applied to bandit settings, such as the logistic bandit (Filippi et al., 2010) and the multinomial logistic bandit (Amani and Thrampoulidis, 2021). For example, in the multinomial logistic bandit problem, MNL-UCB needs to maintain a high-dimensional Hessian matrix, which incurs high computational costs. By applying the sketching technique, we can reduce its space and time complexities.
>
> **Reference:**
>
> Jiawei Zhao, Zhenyu Zhang, Beidi Chen, Zhangyang Wang, Anima Anandkumar, Yuandong Tian. GaLore: Memory-Efficient LLM Training by Gradient Low-Rank Projection. ICML, 2024
>
> Sarah Filippi, Olivier Cappe, Aurélien Garivier, Csaba Szepesvári. Parametric bandits: The generalized linear case. NeurIPS, 2010.
>
> Sanae Amani and Christos Thrampoulidis. UCB-based algorithms for multinomial logistic regression bandits. NeurIPS, 2021.

---

### Decision · Program_Chairs · 2025-05-01

**Decision:**

Accept (poster)

**Comment:**

The paper proposes a new algorithm called Follow-the-Sketchy-Leader (FTSL), which is an adaptive subgradient method for online convex optimization. It shows that the algorithm improves upon the existing regret bounds and time complexities. While the reviewers appreciate the new results, they also indicate that the contributions need to be better articulated, especially since the technical development seems to draw heavily on existing techniques. Also, as one reviewer pointed out, the rigor of the presentation needs to be improved. The author(s) should take the reviewers' comments into account when revising the paper.